# Zinc isotopic evidence for recycled carbonate in the deep mantle

Xiao-Yu Zhang[1], Li-Hui Chen [2,1] ✉, Xiao-Jun Wang[2], Takeshi Hanyu [3], Albrecht W. Hofmann [4], Tsuyoshi Komiya [5], Kentaro Nakamura[6], Yasuhiro Kato [6], Gang Zeng[1], Wen-Xian Gou[1] & Wei-Qiang Li [1]

Carbonate, the major carbon reservoir on Earth's surface, can enter into the mantle by subduction. However, evidence for recycled surficial carbonates in the deep mantle is still scarce. Ocean island basalts from Cook-Austral islands and St. Helena Island, widely called HIMU basalts because of their high $\mu = {}^{238}U/{}^{204}Pb$ sources, are thought to be fed by mantle plumes originating in the lower mantle. Here we report exceptionally high $\delta^{66}Zn$ values ($\delta^{66}Zn = 0.38 \pm 0.03‰$) of these HIMU lavas relative to most published data for oceanic basalts ($\delta^{66}Zn = 0.31 \pm 0.10‰$), which requires a source contributed by isotopically heavy recycled surficial carbonates. During subduction of the oceanic lithosphere, melting of mixed surficial carbonates and basaltic crust in the deep mantle generates carbonatite melts, which metasomatizes the nearby mantle and the resultant carbonated mantle ultimately evolves into a high-$\delta^{66}Zn$ HIMU source. High-$\delta^{66}Zn$ signatures of HIMU basalts, therefore, demonstrate that carbonates can be transported into Earth's deep mantle.

Carbon on Earth's surface, stored as organic carbon and carbonates, can be transported into the Earth's interior by subducting oceanic slabs[1]. Subducted carbonates account for about 70% of the total carbon input[1], part of them (e.g., 40%–65% in ref. 2; 20%–80% in ref. 3) can be removed from the subducting slab by dissolution, melting and metamorphic decarbonation in the shallow upper mantle, while the remainder may enter further into the deeper part of the mantle[1,4,5]. However, traces of recycled surficial carbonates in the deep mantle (i.e., the mantle transition zone and the lower mantle) are still rarely captured thus far. Although studies on superdeep diamonds and their high-pressure mineral inclusions (e.g., Ca(Ti, Si)O₃-perovskite and calcic-majoritic garnet) have suggested the deep cycling of surficial carbon into the lower mantle[6], the low-$\delta^{13}C$ signatures (mean $\delta^{13}C = -28.3‰$ to $-4.8‰$[7,8], mostly < −20‰) of those diamonds are mainly inherited from subducted organic carbon ($\delta^{13}C_{organic\ carbon} \approx -57‰$ to $-28‰$[9]) rather than sedimentary carbonates ($\delta^{13}C \approx 0‰$[10,11]). Recently reported high $\delta^{18}O$ values (+9.1‰ to +10.5‰) of some high-

pressure mineral inclusions in superdeep diamonds, together with the host diamonds' carbon and nitrogen isotopic compositions, were regarded as evidence for subduction of carbonated igneous oceanic crust, rather than sedimentary carbonates, into transition-zone depths[12]. Therefore, convincing isotopic evidence is still lacking to distinguish the signature of sedimentary carbonates that transported into the deep mantle.

The HIMU (high $\mu = {}^{238}U/{}^{204}Pb$) component, one of the end-members in the deep mantle, is defined by ocean island basalts (OIBs) from the Cook-Austral Islands (Mangaia, Tubuai, Old Rurutu, Old Raivavae) and St. Helena Island, which have extremely radiogenic Pb isotopic compositions ($^{206}Pb/^{204}Pb > 20.5$, ref. 13) and deep mantle plume origin[14]. There are several lines of evidence which support the interpretation that the HIMU source should be genetically linked to a carbonate component[15–18]: (1) high-CaO/Al₂O₃ and low-SiO₂ signatures of HIMU basalts coincide with those of experimental melts derived from carbonate-bearing peridotite or pyroxenite[19]; (2) the whole-rock

[1]School of Earth Sciences and Engineering, State Key Laboratory for Mineral Deposits Research, Nanjing University, Nanjing 210023, China. [2]Department of Geology, State Key Laboratory of Continental Dynamics, Northwest University, Xi'an 710069, China. [3]Research Institute for Marine Geodynamics, Japan Agency for Marine-Earth Science and Technology, Yokosuka 237-0061, Japan. [4]Abteilung Klimageochemie, Max-Planck-Institut für Chemie, D-55128 Mainz, Germany. [5]Department of Earth Science and Astronomy, The University of Tokyo, Tokyo 153-8902, Japan. [6]Department of Systems Innovation, School of Engineering, The University of Tokyo, Bunkyo-ku, Tokyo 113-8656, Japan. ✉e-mail: chenlh@nwu.edu.cn

trace element patterns of HIMU lavas show similarity to those of high Mg-carbonatitic liquids encapsulated in diamonds[16]; (3) carbonate globules are observed in melt inclusions from Mangaia lavas, supporting a primary $CO_2$-rich magma in Cook-Austral chains[18]; (4) olivine phenocrysts in HIMU lavas have high Ca contents (up to 3,200 p.p.m.), suggesting that a mantle source which have been metasomatized by carbonatitic fluids[16]; and (5) carbonated peridotite xenoliths found in Tubuai (Austral Islands) demonstrate that the lithospheric mantle has been metasomatized by carbonatitic liquids from the mantle plume[17]. However, carbonate in the deep Earth might be introduced not only by recycled surficial carbonates, but also by oxidation of reduced primordial carbon at the base of the mantle transition zone[20]. Thus far, convincing isotopic tracer for recycled carbonates can further strengthen the genetic link between the carbonate component in the HIMU mantle sources and the recycled surficial carbonates.

Zinc isotopes appear to discriminate between the mantle and crustal materials. The uniquely high-$\delta^{66}Zn$ signature of surficial carbonates clearly distinguishes these rocks from normal mantle compositions and mid-ocean ridge basalts (MORB) (Fig. 1). Moreover, this tracer shows only very limited variations (<0.1‰) during crystal fractionation and partial melting[21,22]. Thus, zinc isotopic studies of basalts provide powerful tools for identifying recycled surficial carbonates in mantle sources[23,24].

Here, we perform high-precision Zn isotopic analyses on classic HIMU OIBs from the Cook-Austral Islands and St. Helena Island, Archean altered oceanic crust and carbonates. For comparison with these HIMU OIBs, we also report Zn isotopic compositions of basalts from the Louisville seamount chain, which have FOZO (FOcus ZOne, ref. 25)-type Sr-Nd-Pb isotopic compositions ($^{87}Sr/^{86}Sr = 0.70360–0.70373$, $\varepsilon_{Nd} = 3.13–4.25$ and $^{206}Pb/^{204}Pb = 18.4–19.2$, ref. 26). Our results provide clear isotopic evidence for a surficial origin of the carbonate component in the HIMU sources, and further elucidate the characteristics and origin of the famous HIMU component, utilizing a full data set (major and trace elements, radiogenic isotopes and stable Zn isotopes) of a same batch of OIB samples (Supplementary information, Note 1, Supplementary Table 1). Our new observations confirm that recycled surficial carbonates have played a

fundamental role in generating the HIMU component in the deep mantle.

## Results

### High $\delta^{66}Zn$ signature of the HIMU Mantle Sources

HIMU lavas from St. Helena Island and the Cook-Austral Islands have similar $\delta^{66}Zn$ values ranging from 0.36‰ to 0.44‰ and from 0.35‰ to 0.41‰, respectively (Fig. 2a and Supplementary Table 1.1). It is striking that $\delta^{66}Zn$ values of the HIMU lavas are distinctly higher than those of normal peridotitic mantle ($\delta^{66}Zn = 0.16 \pm 0.06$‰, ref. 27 or $\delta^{66}Zn =$

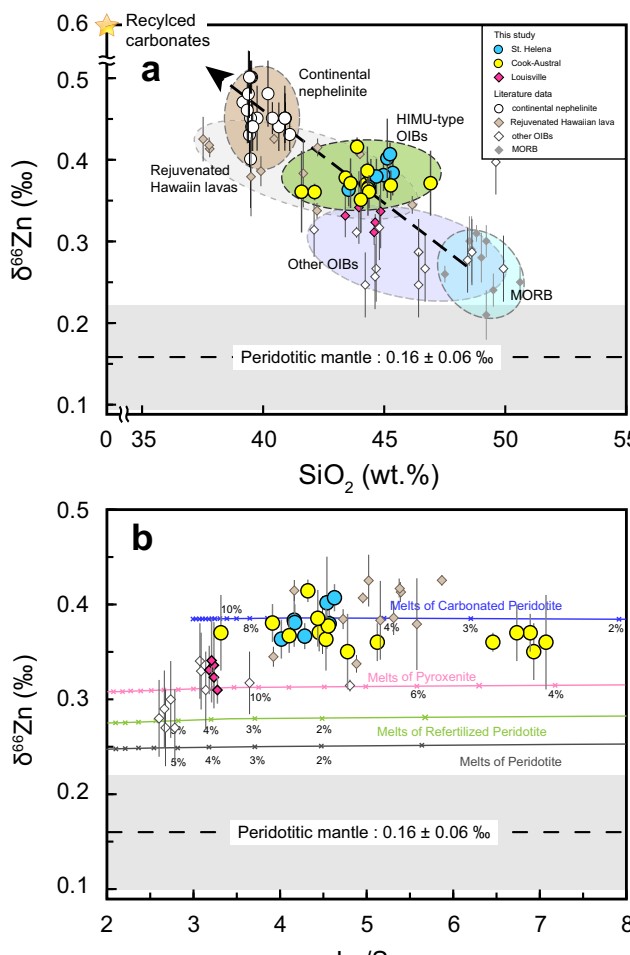

**Fig. 2 | Variations in $\delta^{66}Zn$ versus $SiO_2$ (a) and La/Sm (b) for HIMU basalts.** All samples in **a** have high MgO contents (MgO>8 wt.%). Therefore, all samples shown in **a** are less-evolved samples and are suitable for discussion about the nature of its mantle source. Classic HIMU OIBs are shown by yellow (Cook-Austral samples) and blue circles (St. Helenea samples). Zinc isotopes and other geochemical data (major and trace elements) of HIMU basalts can be found in Supplementary information, Note 1, Supplementary Table 1.1 and Supplementary Table 7. Estimated $\delta^{66}Zn$ values of the peridotitic mantle are taken from ref. 27. $\delta^{66}Zn$ values of samples from continental nephelinite, MORB and other OIBs are also shown (Data are from refs. 21–23, 29–31). The black arrow in **a** points to the direction of increasing contribution of recycled carbonates. Error bars on $\delta^{66}Zn$ represent 2 standard deviations (2 SD) uncertainties. The gray line in **b** represents model calculations for melts of peridotite. The green line represents model calculations for melts of refertilized peridotite. Silicic melts released from recycled oceanic crust metasomatize normal peridotite and therefore produce such refertilized peridotite, as mentioned in Herzberg et al.[41]. The pink and blue lines represent model calculations for melts of pyroxenite and carbonated peridotite, respectively. Inset numbers indicate the degrees of partial melting. See Supplementary information, Note 4 for more details about model calculations of mantle partial melting. MORB = Mid-Ocean Ridge Basalt; OIB = Ocean Island Basalt; HIMU = high μ ($\mu = ^{238}U/^{204}Pb$).

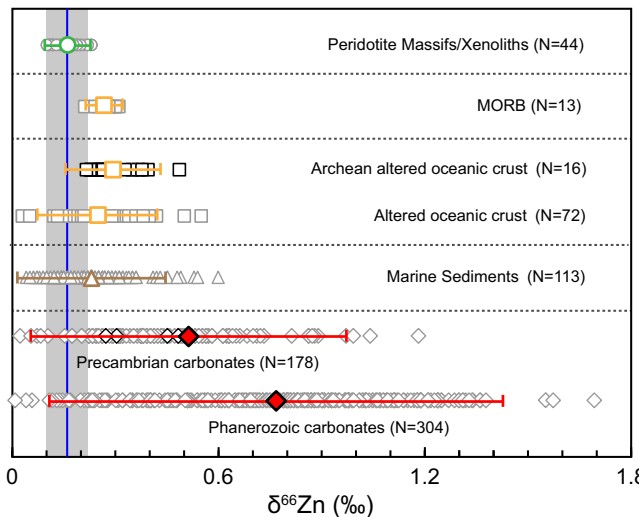

**Fig. 1 | Zinc isotopic compositions of mantle peridotite and several crustal reservoirs.** The blue line marks the average $\delta^{66}Zn$ value of the peridotitic mantle ($\delta^{66}Zn = 0.16$‰, ref. 27) and the vertical gray field marks the $\delta^{66}Zn$ range of the peridotitic mantle ($\delta^{66}Zn = 0.10$‰-0.22‰, ref. 27). The number of samples (N), average $\delta^{66}Zn$ values and 2 standard deviations (2 SD) are given. Reference data are shown as symbols with gray rims, and data of this study are shown as symbols with black rims. See Supplementary Information, Note 6 for data sources. MORB = Mid-Ocean Ridge Basalt.

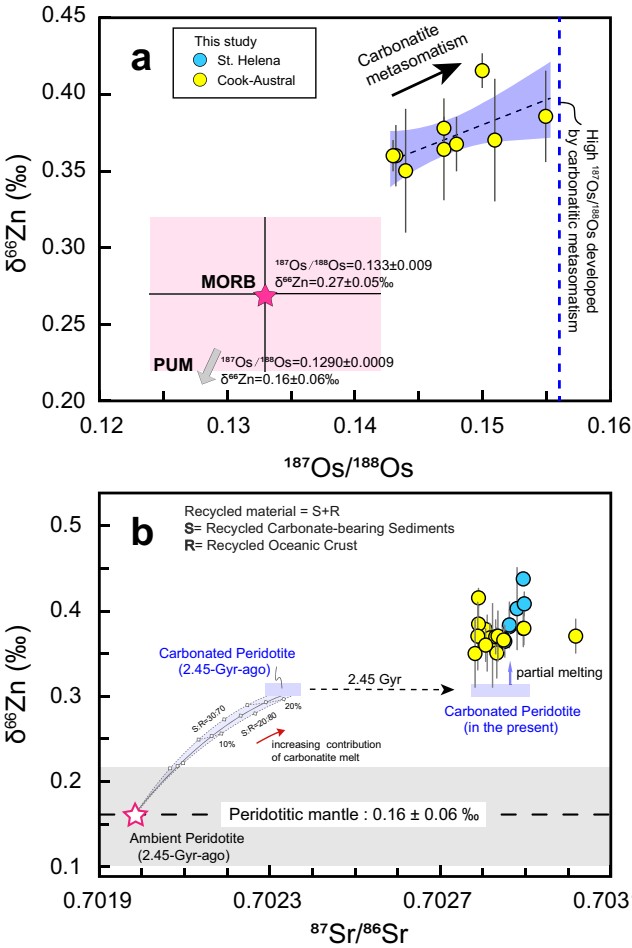

**Fig. 3 | Variations in δ⁶⁶Zn versus age-corrected ¹⁸⁷Os/¹⁸⁸Os values (a) and ⁸⁷Sr/⁸⁶Sr values (b) of HIMU basalts.** In Figure 3a, Age-corrected ¹⁸⁷Os/¹⁸⁸Os values for Cook-Austral samples are from ref. 35 (Supplementary Table 7). The blue line represents the calculated high ¹⁸⁷Os/¹⁸⁸Os ratio (0.156, ref. 16) for carbonatite-metasomatized peridotite mantle[16]. Error bars on δ⁶⁶Zn represent 2 standard deviations (2 SD) uncertainties. In Figure 3b, δ⁶⁶Zn value of peridotitic mantle is given by ref. 27. The black dashed line marks the average value and the gray field highlights the uncertainty range. All data source for factors of calculations in **a** and **b** are given in Supplementary Information, Note 6. MORB = Mid-Ocean Ridge Basalt. PUM = Primitive Upper Mantle.

variation of St. Helena samples is generated by fractionation of mafic minerals (Supplementary Figure 1 and Supplementary Figure 2). Nevertheless, it is worth noting that even if the variation of δ⁶⁶Zn values and MgO contents of St. Helena samples is consistent with the trend affected by fractionation of mafic minerals, the theoretical initial δ⁶⁶Zn value of primary magma of St. Helena should be as high as 0.37‰. Moreover, the most primitive St. Helena sample with the highest MgO (15.72wt.%) does have an elevated δ⁶⁶Zn value (0.36‰), which is still distinctively higher than those of MORBs (0.27 ± 0.05‰, refs. 21, 29) and other OIBs (0.33 ± 0.09‰, refs. 21, 22, 30, 31). Therefore, the high-δ⁶⁶Zn signature of HIMU OIBs from St. Helena Island mainly reflects the primary magma composition, rather than effect of fractional crystallization. For Cook-Austral samples, their high MgO contents (MgO>8wt.%) and the lack of correlation between MgO and δ⁶⁶Zn values rule out fractional crystallization as a primary control on the Zn isotopic compositions (Supplementary Figure 2).

Partial melting could also cause minor fractionation of Zn isotopes[27]. Our quantitative modelling shows that maximum δ⁶⁶Zn values of melts produced by peridotite and pyroxenite partial melting are 0.25‰ and 0.32‰, respectively, and are obviously lower than those of HIMU lavas (Fig. 2b, Supplementary Figure 3 and Supplementary information, Note 4). Melting under garnet stability field can also hardly explain the heavy Zn isotopic compositions of HIMU OIBs[27,28]. Since OIBs are from similar depths where garnet is the residue phase, the pressure effect, if it did exist, would have a similar influence on both HIMU lavas and other OIBs, which is at odds with the observation that HIMU lavas have higher δ⁶⁶Zn values relative to most other OIBs (Fig. 2a). Hence, the high-δ⁶⁶Zn signature of the HIMU basalts cannot be generated solely by melting of normal peridotite or pyroxenite mantle sources, but requires the HIMU mantle sources with high-δ⁶⁶Zn signature.

## Origin of the HIMU component

The recycled ancient oceanic crust has been widely regarded as the origin for the HIMU component[34–40]. Hydrothermal alteration and dehydration processes of subducting crust can induce strong fractionation of U, Th and Pb, so as to attain the high U/Pb and Th/Pb ratios[38,39]. Such high U/Pb and Th/Pb ratios can generate the extremely radiogenic Pb isotopic compositions in HIMU mantle source with long-term isolation[38,39]. The relative enrichment of Th and U in the recycled ancient oceanic crust can also account for the low-³He/⁴He signature of HIMU lavas, because α decay of U and Th will increase ⁴He contents and lower ³He/⁴He ratios[34,37,40]. However, olivine phenocrysts of HIMU lavas have low Ni contents and high Mn/Fe ratios (0.013–0.017)[16,41], which is consistent with peridotitic sources rather than pyroxenite sources[16]. Herzberg et al.[41] therefore proposed a "phantom Archean crust" in the HIMU source: the lithological identity (eclogite) of recycled mafic crust no longer exists in the source of HIMU basalts, but its released silicic melts modified the chemistry of the surrounding mantle and yielded a refertilized peridotite[41]. However, the calculated maximum δ⁶⁶Zn values of melts produced by such a carbonate-free refertilized peridotite is 0.28‰ only (Fig. 2b, Supplementary Figure 3 and Supplementary information, Note 4). Thus, this "phantom" model, involving pure silicate liquids, will not generate the elevated δ⁶⁶Zn signatures (0.38 ± 0.03‰) observed in HIMU basalts (see Fig. 2b and Supplementary information, Note 4).

An alternative explanation proposed is that the HIMU source is a peridotite which has been metasomatized by carbonatite melts[16–18,42,43], which has been supported by previous observations of olivine phenocrysts with high CaO contents[16,32,41] and carbonate globules in their melt inclusions[18]. Such metasomatized peridotite is enriched by carbonatitic fluids with low SiO₂, Al₂O₃ and high CaO contents[16,41]. In addition, the similarity in trace-element patterns, e.g., depletions in K, Rb and Pb relative to other incompatible elements, between HIMU lavas and high-Mg carbonatitic liquids encapsulated in diamonds also

0.20 ± 0.03‰, ref. 28) (Fig. 2a). Notably, δ⁶⁶Zn values of HIMU OIBs from the Cook-Austral Islands and the St. Helena Island are the highest δ⁶⁶Zn values so far recorded in oceanic basalts (δ⁶⁶Zn = 0.31 ± 0.10‰, calculated from refs. 21, 22, 29–31), higher than MORBs (δ⁶⁶Zn = 0.27 ± 0.05‰, refs. 21, 29) and other OIBs from Louisville (δ⁶⁶Zn = 0.33 ± 0.01‰, this study), Crozet, Hawaii and Iceland (δ⁶⁶Zn = 0.28 ± 0.08‰, refs. 21, 22, 30). In addition, these high δ⁶⁶Zn values are close to those of the rejuvenated-stage lavas from Kaua'I, Hawaii islands (δ⁶⁶Zn = 0.38 ± 0.05‰, ref. 31) and silica-undersaturated nephelinites from eastern China (δ⁶⁶Zn = 0.45 ± 0.05‰, ref. 23).

These HIMU OIBs have experienced fractionation of olivine and clinopyroxene because their MgO contents vary in a broad range and correlate positively with CaO/Al₂O₃ ratios[32,33]. Fractionation of olivine (δ⁶⁶Zn = 0.15‰, α$_{ol-melt}$ = 0.99990) and clinopyroxene (δ⁶⁶Zn = 0.16‰, α$_{cpx-melt}$ = 0.99995) can slightly increase δ⁶⁶Zn values (<0.1‰) of the evolved melts[21,22,28]. Thus there is a positive correlation of SiO₂ versus δ⁶⁶Zn and a negative correlation of MgO versus δ⁶⁶Zn for St. Helena samples (Supplementary Figure 1 and Supplementary Figure 2). These observations, together with the calculated effect of fractional crystallization (Supplementary information, Note 3), suggest that δ⁶⁶Zn

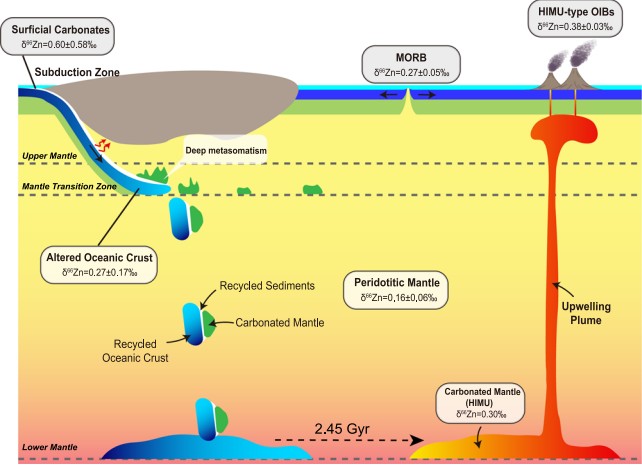

**Fig. 4 | Cartoon depicting the proposed genesis of HIMU component.** Estimated average Zn isotopic compositions for various reservoirs are also shown. The average values for MORB, altered oceanic crust and sedimentary carbonates are calculated by data measured in this study and data from reference. All data source for Fig. 4 is given in Supplementary Information, Note 6. The average value for carbonated mantle is assumed in this study. Estimated $\delta^{66}Zn$ value of peridotitic mantle is taken from ref. 27. MORB = Mid-Ocean Ridge Basalt; OIB = Ocean Island Basalt; HIMU = high $\mu$ ($\mu = {}^{238}U/{}^{204}Pb$).

suggests the involvement of carbonatite metasomatism in the HIMU source[16].

Figure 2b shows the melting curve for such a carbonatite-metasomatized mantle peridotite with $\delta^{66}Zn$ of 0.30‰. The $\delta^{66}Zn$ values of partial melts generated from such a source reach 0.39‰ (Fig. 2b and Supplementary Figure 3), which matches the Zn isotopic composition of the HIMU basalts (0.38 ± 0.03‰ on average). Thus, melting of peridotite metasomatized by such carbonatite liquids can generate the HIMU lavas with characteristically heavy Zn isotopic signature.

**Nature of the metasomatic carbonatite liquids**

The question then arises whether the carbonatite liquids were derived from recycled carbonates or, as recently argued by Sun et al.[20], whether they were generated by oxidation of diamond in the mantle transition zone. The breakdown of bridgmanite in the mantle transition zone, the main mineral phase of the lower mantle, releases $Fe^{3+}$, increasing the redox state. Sun et al.[20] therefore proposed that carbon(ate) in the mantle transition zone is derived from the primitive mantle rather than from recycled carbonates. However, the high $\delta^{66}Zn$ values of HIMU basalts are not consistent with such an origin. The degree of stable isotope fractionation is inversely proportional to temperature[44]. Such carbonates produced at high temperature in the mantle transition zone (e.g. 1800 K) are not expected to show significant isotopic fractionation relative to the normal mantle, and liquids released from such a carbonate-bearing mantle should have similar $\delta^{66}Zn$ values compared to the silicate mantle.

Another important observation is that high ${}^{187}Os/{}^{188}Os$ and $\delta^{66}Zn$ signature of HIMU samples from the Cook-Austral islands are positively correlated (Fig. 3a). Oceanic and continental crustal materials have much higher Re/Os ratios than peridotites[45]. Since Re is more incompatible than Os and is enriched in metasomatic liquids, metasomatism originating from recycled crustal materials can increase Re/Os ratios of mantle peridotites. The β decay of ${}^{187}Re$ to ${}^{187}Os$ produces measurable variations in the abundance of ${}^{187}Os$ in the mantle. If such a metasomatic event occurred very early, e.g. in the late Archean to early Proterozoic[16,46], a high ${}^{187}Os/{}^{188}Os$ HIMU reservoir can be generated by the long-term radiogenic accumulation of ${}^{187}Os$[16]. Therefore, elevated ${}^{187}Os/{}^{188}Os$ ratios reflect the influence of the recycling of crustal

materials[35,36]. The coupling of Os and Zn isotopic compositions suggests that such a high $\delta^{66}Zn$ signature is related to recycled crustal materials and, similar to radiogenic Os isotopes, is an intrinsic signature of the HIMU component. We infer that the elevated $\delta^{66}Zn$ signature of the HIMU basalts is caused by the addition of a recycled surficial component with heavy Zn isotopic compositions.

Surficial carbonates, which are formed at low temperature, stand out by their high $\delta^{66}Zn$ values (up to 1.34‰, Fig. 1). Some kinds of Mg-rich carbonates have significantly high Zn contents (e.g., 147p.p.m in dolomite and 449p.p.m in magnesite)[47]. Such Mg-rich carbonates are more stable in the subduction zone[3,48], and a considerable proportion of them has the potential to be retained in subducted slabs and finally introduced into the deep mantle, e.g., the mantle transition zone (410–660 km)[48,49]. Consequently, recycled carbonate, along with recycled oceanic crust, is a suitable candidate to produce metasomatic carbonatite melts (with variable proportions of silicate composition), which inherit the high $\delta^{66}Zn$ signature. High-pressure experiments suggest that subducted carbonate-bearing sediments/oceanic crust will partially melt at the top of the mantle transition zone and release carbonatite melts[50,51]. As an incompatible element, zinc will be preferentially enriched in the carbonatite melts. Ambient peridotite has much lower Zn contents (~26p.p.m, refs. 21, 29) and light Zn isotopic compositions ($\delta^{66}Zn = 0.16 ± 0.06$‰, ref. 27). Thus, carbonatite liquids released from recycled carbonates can exert considerable influence over Zn isotopic compositions on mantle peridotite, generating the metasomatized peridotite with elevated $\delta^{66}Zn$ values. Our modeling (Supplementary information, Note 5) suggests that only 6–12% of originally subducted carbonates are required to generate a HIMU source with $\delta^{66}Zn$ of 0.30‰. Because of the residue of liebermannite, carbonatite melt generated in the mantle transition zone depletes in Rb and thus has extremely low Rb/Sr ratios[52]. The ${}^{87}Rb/{}^{86}Sr$ ratio of carbonatite melt is as low as 0.002[6]. Therefore, carbonatite melt with low Rb/Sr ratios cannot increase Rb/Sr ratio of peridotite mantle in the HIMU source. A low ${}^{87}Rb/{}^{86}Sr$ HIMU reservoir cannot generate high ${}^{87}Sr$ contents even by the long-term radiogenic accumulation. For a metasomatized event at 2.45Ga[46], only 21%–25% carbonatite melt with low Rb/Sr ratios is required to produce ${}^{87}Sr/{}^{86}Sr$ values of HIMU source (Fig. 3b), and such proportion of metasomatized melt is consistent with estimates based on radiogenic Os isotopes (18–30%) given by Weiss et al.[16] and Pb isotopes (25%) given by Chauvel et al.[39]. Melting of such carbonated peridotite can therefore explain the high-$\delta^{66}Zn$ signature, high ${}^{187}Os/{}^{188}Os$ and low ${}^{87}Sr/{}^{86}Sr$ values of the HIMU basalts.

Furthermore, an inverse relationship between $SiO_2$ contents and $\delta^{66}Zn$ values can be found in Fig. 2a. All samples in Fig. 2a have only experienced fractional crystallization of a minor or small amount of mafic minerals (e.g. olivine and clinopyroxene), so the wide range of $SiO_2$ contents (39.5–50.6wt.%) from MORB to continental nephelinites can hardly be attributed to crystal fractionation. Besides, experimental melts of garnet peridotite have a narrow range of $SiO_2$ contents (45–48wt.%), even though they were produced by various degrees of partial melting at different high pressure (3–7 GPa)[53]. Therefore, partial melting has limited effect on the significant variation in $SiO_2$ contents (39.2–50.6wt.%) shown in Fig. 2a. By contrast, it has long been known that $SiO_2$ contents of melts produced by melting of carbonated peridotite decrease significantly with increasing dissolved $CO_2$ concentration in the melts[19,54], resulting in a negative correlation between $SiO_2$ contents and $\delta^{66}Zn$ values (Fig. 2a). The greater the contribution of such carbonatite liquids in the source, the lower the $SiO_2$ contents and higher the $\delta^{66}Zn$ values in the lavas will be, generating a negative correlation on the $SiO_2$-$\delta^{66}Zn$ plot (Fig. 2a). The continental nephelinites from eastern China, which have been regarded as direct melts derived from recycled carbonate-bearing crustal materials[52], display the highest $\delta^{66}Zn$ values (0.40‰–0.50‰)[23]. The obviously lower $\delta^{66}Zn$ values (<0.3‰) of MORBs and some OIBs relative to those of HIMU lavas and

continental nephelinites can be explained by melting of carbonate-free mantle sources.

In conclusion, our study confirms that not only basaltic crust but also recycled marine carbonates have played key roles in the generation of HIMU basalts. Melting of subducted basaltic crust together with recycled carbonates would produce carbonatite liquids (with variable proportion of silicate composition), which inherit their high-$\delta^{66}$Zn signature from recycled carbonates, and also inherit high Re/Os, U/Pb, Th/Pb and (U + Th)/³He ratios, as well as high FeO and $TiO_2$ contents, from the subducted basaltic crust[34–40,55,56]. After such a carbonatite melt has metasomatized normal peridotite and thus produced a potential HIMU source, with high-$\delta^{66}$Zn signature, in the deep mantle. A long-term (2 to 3 billion year) isolation[39,46] is required to develop the characteristic isotopic signatures of HIMU basalts, including the extremely high $^{206}Pb/^{204}Pb$, low ³He/⁴He and high $^{187}Os/^{188}Os$ ratios. But such long-term isolation is not essential for all types of HIMU basalts. For example, the source of the recently discovered HIMU basalts from a Bermuda drill core was generated less than 650 million years ago and may in fact have been formed in the mantle transition zone immediately prior to their eruption about 30 million years ago[57].

### HIMU as the deep carbon reservoir

The cartoon shown in Fig. 4 summarizes the implications of our findings for the crustal recycling and the generation of HIMU basalts: (1) Subducted oceanic crust and carbonates undergo partial melting and release carbonatite melts, which metasomatize the overlying peridotite mantle. The metasomatized peridotite "reservoirs" are shown in green in Fig. 4. This process is likely to take place at depths exceeding 300 km[57], i.e. in the pressure range where liebermannite, formerly called K-hollandite[58], is stable as a residual phase in the subducted crust. This mineral preferentially retains Pb but releases U and Th[59], and it thus imparts a high (U,Th)/Pb ratio on the melt and on the metasomatized peridotite. This process can thus generate a carbonated mantle with an elevated U/Pb ratio. While we cannot rule out the model of Weiss et al.[16], whereby the carbonate metasomatism infiltrates the base of the subcontinental lithosphere, which is significantly shallower (≤ 200 km), we note that liebermannite (which has high partition coefficients for Rb, K, Ba) is not stable in the subcontinental lithosphere, consequently that model does not provide a compelling mechanism for generating the extremely low Ba/Th ratios (with a mean of 60.8) that are characteristic of HIMU basalts. (2) In order to develop the highly radiogenic Pb isotopes of HIMU basalts such as those found on Mangaia or St. Helena, the metasomatized assemblage must be stored in the deep mantle for a period of the order of 2 billion years. We suggest that this is accomplished by an entrainment process, whereby the subducted slab entering the lower mantle drags the metasomatized peridotite toward the base of the mantle (shown in Fig. 4 as green slabs attached to blue lithosphere). After the appropriate time delay, during which the HIMU-type radiogenic isotopic signatures developed, the metasomatized peridotite is incorporated into deep-seated mantle plumes, which rise to the shallow asthenospheric mantle and produce HIMU-type lavas.

Evidence for subducted carbonate, using Mg isotopes, has previously been identified also in another isotopic endmember composition of OIBs, namely the EM1 (EMI = "Enriched Mantle-1") basalts from Pitcairn[60]. Chemically and isotopically, such EM1 sources are complementary to HIMU sources. Subducted marine carbonates thus contribute a critical "ingredient" to generating both types of sources through the action of carbonate melting in the presence of liebermannite (K-hollandite). The carbonate melt creates the HIMU source by infiltrating the overlying peridotites, whereas the liebermannite-bearing residue concentrates the elements K, Rb, Ba, and Pb, which leads to the characteristically unradiogenic Pb isotopic compositions as well as high K/U and low U/Pb ratios seen in EM1 basalts. Meanwhile, the complementary HIMU peridotite assemblage

displays low K/U and high U/Pb ratios. Both of these extreme types of OIB sources are found in mantle plumes occurring both the Pacific and Atlantic ocean basins. Furthermore, recent investigations have shown that HIMU mantle domains occur also at many locations worldwide, beneath continental (Archean cratonic, e.g. Kaapvaal craton; rift volcano, e.g. East African Rift) and oceanic plates (oceanic islands/seamounts, e.g. Austral-Cook Islands; oceanic plateaus, e.g. Manihiki Plateau)[42]. Thus, subducted marine carbonates have left their traces in many places and a significant portion of the $CO_2$ outflux via intraplate volcanism may be derived from recycled carbonate in the deep mantle.

## Methods
### Zn isotopes

Purification and isotopic analysis of Zn were performed at the Key Laboratory of Surficial Geochemistry, Ministry of Education, School of Earth Sciences and Engineering in Nanjing University, China, following previous procedures established by ref. 61. Approximately 0.1–0.2 g of each sample and USGS (the United States Geological Survey) reference materials were completely digested by concentrated double-distilled acids in the Savillex screw-top beakers, according to the following sequence: (i) digestion in HF-HNO₃ (1:2, vol/vol), (ii) digestion in HCl-HNO₃ (3:1, vol/vol), and (iii) digestion in HCl (3 ml, three times). Subsequently, the clear solutions were dried and dissolved in 2 M HCl for ion exchange purification. Zn purification was achieved by anion exchange process in columns loaded with Bio-Rad AGMP-1M (100–200 mesh) resin, following established procedures[61]. Zinc after purification was re-dissolved in 0.05 M HNO₃ for isotopic analysis. At the end of the procedure, the Zn yield was ≥99.5% for all analyzed samples, including reference materials and unknown samples. The total procedural Zn blank ranged from 2 ng to 20 ng.

Zinc isotopic compositions were measured on a Neptune Plus (Thermo Fisher Scientific) MC-ICP-MS. Mass bias and instrument drift was corrected by using combined sample-standard bracketing (SSB) and empirical external normalization (EEN) method with a Cu NIST647 doping. An in-house High Purity Standard (HPS) Zn solution was used as the bracketing standard for measurement. Each sample was measured at least three times. Zn isotopic compositions are expressed relative to the JMC Lyon standard as:

$$\delta^{66}Zn = [(^{66}Zn/^{64}Zn)_{Sample}/(^{66}Zn/^{64}Zn)_{JMCLyon} - 1] \times 1000 \quad (1)$$

HPS has a $\delta^{66}$Zn value of −0.06 ± 0.04‰ relative to the JMC Lyon standard[61,62]. Therefore, $\delta^{66}Zn = \delta^{66}Zn_{HPS} - 0.06$. Based on the repeated analysis of standard solutions and various rock reference materials, the long-term external reproducibility (2 standard deviations, 2 SD) is better than ±0.04‰ for $\delta^{66}$Zn[61]. Analytical results for USGS rock reference materials (BCR-2, BHVO-2, BIR-1, AGV-2 and JDo-1) in this study are consistent with those of previous studies (Supplementary Table 2). In a plot of $\delta^{66}$Zn vs. $\delta^{68}$Zn, all measured samples fall along the expected mass-dependent fractionation line for Zn isotopes (Supplementary Figure 4).

## Data availability

The data generated in this study are available in Supplementary Tables and also from the corresponding author upon reasonable request. Referenced data supporting the findings of this study are available in either the published works cited or GEOROC database (http://georoc.mpch-mainz.gwdg.de/georoc).

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

## Acknowledgements

We appreciate laboratory or technical support from Gao-Jun Li, Wei Li, Jin-Hui Yang, Yue-Heng Yang and Lie-Wen Xie. Hiroshi Kawabata, Tao Li and Mao-Yu Wang is acknowledged for sample preparation. Saskia Erdmann polished the early version of this manuscript. This study was financially supported by the National Natural Science Foundation of China (Grant. 42130310 to L.-H.C., and Grant.41973001 to X.-J.W.).

## Author contributions

L.-H.C. conceived the project. X.-Y.Z., L.-H.C., X.-J.W. and A.W.H. wrote the manuscript. X.-Y.Z., W.-X.G. and W.-Q.L. measured Zn isotopes. T. H., T.K., K.N. and Y.K. provided the samples. X.-Y.Z., L.-H.C., X.-J.W., A.W.H., T. H., and G.Z. analyzed data. All authors contributed to the preparation of the final manuscript.

## Competing interests

The authors declare no competing interests.
