## [Peer Review File · Nature Communications]

Zinc isotopic evidence for recycled carbonate in the deep mantleReviewer #1 (Remarks to Author) :

Review

Manuscript Number: NCOMMS-21-23139-T

Title: Zinc isotopic evidence for recycled carbonate in the deep mantle

Authors: Xiao-Yu Zhang, Li-Hui Chen, Xiao-Jun Wang, Takeshi Hanyu, Albrecht W. Hofmann, Tsuyoshi Komiya, Kentaro Nakamura, Yasuhiro Kato, Gang Zeng, Wen-Xian Gou, Wei-Qiang Li

This paper presents data analyses for the isotopic compositions of Zn of HIMU basalts from Cook-Austral and St. Helena islands. The results presented support previous suggestions that the mantle source of HIMU involves subducted carbonate via carbonatite metasomatism (e.g. Weiss et al., 2016). It is important that the new data will be published as they strengthen our understanding of the evolution of the extreme HIMU mantle component. However, I believe the authors should better explain and demonstrate in the text what is the major and unique contribution of the new data. For example, the authors mention that ‘it is not known how far such carbonate can be transported into the deeper mantle.’ (Line 34-35) and ‘therefore, demonstrate that carbonates can be transported into Earth’s deep mantle and recycled back to...’ (Line 52-53); but data collected on the carbon isotopic composition and inclusions of diamonds originating from the transition zone and lower mantle, indicate the depth to which crustal carbonate is recycled back to the mantle. The model presented in the manuscript explains the Zn isotopic composition of the HIMU source, but does not show or sufficiently discuss it in the context of the other characteristic isotopic compositions of this mantle component (i.e. Sr, Nd, etc. isotopic compositions). In addition, the authors provide a few ‘outstanding’ sentences in the text like ‘Earth’s carbon cycle has played a fundamental role in creating a habitable planet over geological time.’ (Line 54), ‘our findings for Earth’s deep carbon cycle’ (line 304) and ‘Figure 4 Cartoon depicting Earth’s deep carbon cycle and....’, which sound a bit out of the scope of the current manuscript.

In summary, I believe major revisions are needed before this manuscript is considered for publication in Nature Communications.

Detailed comments

Line 61-62: ‘estimates ranging from minor to major’ – these amount estimations do not say much; numbers as tonne per year should be given.

Line 62-63: ‘it is uncertain whether carbonates in subducted slabs can be transported into the deep mantle, i.e., into the mantle transition zone and the lower mantle’ – diamonds from the deep mantle holds direct evidence for the depth to which recycled

carbon is subducted (e.g. Cartigny, Elements, 2005; Thomson et al., CMP, 2014; Smith et al., nature, 2018; Li et al., EPSL, 2019), so this statement should be revised or explained.

Line 90-94: an additional and important supporting evidence for the connection between the HIMU basalts and a carbonated mantle source is the similarity in trace elements between these magmas and deep high-Mg carbonatite encapsulated in diamonds (Weiss et al., nature, 2016).

Line 94-96: this statement is kind of odd regarding the Rohrbach et al. study that is cited, as in this study the mantle domain that experiences carbonatite redox melting within the transition zone is formed at first by subduction.

Line 97-98: 'this says little or nothing about the origin of such carbon(ate).' – what about the other evidence that relates the HIMU source with subduction (e.g. Sulfur isotopes in HIMU basalts), don't they link the carbonate to subduction?

Line 107: for comparison with what?

Line 113-114: 'does not necessarily invalidate previously proposed models for the genesis of HIMU,' – not only that the new observation does not invalidate previously proposed models, rather, I kind of see that the new observation strengthen/support them.

Line 142-143: but few lines above the authors argue that fractional crystallization is not a major factor in the isotopic signature of Zn (lines 137-140); therefore, can this high value be explained differently?

Line 158-171 and 279-300: Mg isotopic signature – The role of these parts in the paper seems slightly weak and unnecessary.

Line 204: in addition, the strong resemblance in trace elements pattern between HIMU basalts and high-Mg carbonatite encapsulated in diamonds, indicates the involvement of carbonatite metasomatism.

Line 253-257: 'Our modeling (Supplementary information, Part 7) suggests that 6–12% of originally subducted carbonates are required to generate a HIMU source with $\delta^{66}\text{Zn}$ of 0.30‰. Melting of such carbonated peridotite can therefore explain the high- $\delta^{66}\text{Zn}$ signature of the HIMU basalts.' – how this modeling coincides with the characteristic HIMU isotopic signature of Sr, Nd, etc.?

Line 269-273: I was always puzzled how a metasomatized carbonated peridotite source can survive for long periods in the convective mantle (especially in its deeper parts), as such a source is susceptible to melting at ambient mantle temperatures. It will be good if the authors can provide some insight on that subject.

Line 285-288: I am not convinced that high Ti/Eu ratios and HIMU-like isotopic signature in xenoliths, by themselves, are strong enough evidence for carbonatitic metasomatism impact. Do the Zealandia xenoliths show any other geochemical features that support such a suggestion?

Line 289-290: reference?

Line 303: ‘summarizes the implication of our findings for the Earth’s deep carbon cycle’ – this statement is highly exaggerating the impact of new data on our understanding of the carbon cycle.

Line 312-315: in the model of Weiss et al. (2016) the high-U/Pb signature is related to the subducting slab (which is part of the model).

Line 319: here again, I am puzzled by how such a metasomatic source with a solidus below the ambient mantle can be preserved for long periods (i.e. billions of years) at the convective mantle.

Line 342-349: sounds like this last part is over the scope of the present manuscript.

Figures:

Figure 1: the line is between 0.1 and 0.2, and not as stated in the captions.

Figure 2: in (a), it would be good to show where the recycled carbonate is situated with respect to the general trend observed for $\delta^{66}\text{Zn}$ vs. SiO_2 .

Figure 3: Is the carbonatite metasomatism arrow direction based on a specific model? or determined based on the position of carbonatitic melts in the $\delta^{66}\text{Zn}$ vs. $^{187}\text{Os}/^{188}\text{Os}$ space? if not, based on what it was determined?

Figure 4: Although many subduction and plume figures have been drawn over the years, this specific one shares too many graphic features and details with the one published by Weiss et al. (2016). Perhaps the authors should consider altering this Figure a bit? or mentioning in the captions something like ‘altered after Weiss et al., 2016’?

Sincerely,

Yaakov Weiss

The Fredy and Nadine Herrmann Institute of Earth Sciences

The Hebrew University, Jerusalem 91904, Israel

Email: yakov.weiss@mail.huji.ac.il

Reviewer #2 (Remarks to the Author):

Review of "Zinc isotopic evidence for recycled carbonate in the deep mantle" by Zhang et al.

Zhang et al. report high precision Zn isotope data for HIMU lavas from St. Helena and Cook-Austral. They found that HIMU lavas have higher $d_{66}\text{Zn}$ than other OIBs, MORB, and peridotites, which they attributed to a signature of recycled carbonate in the HIMU mantle sources. The data are important and should be published. The interpretation can be improved.

My major concern is whether high $d_{66}\text{Zn}$ is a distinctive SOURCE signature of HIMU. The authors did discuss and evaluate the effects of crystal fractionation and partial melting, and concluded that their effects were too small to explain the observed high $d_{66}\text{Zn}$ in HIMU lavas. However, the discussion is not complete.

1. Figure S2 shows all published data together with new data from this study. It is obvious that some published non-HIMU OIBs (Kilauea Iki lavas from Hawaii and Hekla lavas from Iceland from Chen et al., 2013) have $d_{66}\text{Zn}$ as high as HIMU lavas. High $d_{66}\text{Zn}$ in some low MgO Kilauea Iki and Hekla lavas may be a result of extensive crystal fractionation. Hence it is important to group them together to show the effect of crystal fractionation. Crystal fractionation effect is also observed in St Helena lavas, but not obvious in Cook-Austral lavas. – Why?

2. Role of sulfide during crystal fractionation. Sulfide inclusions in olivines have been reported in HIMU lavas (Cabral et al., 2013 Nature; Delavault et al., 2016 PNAS; Dottin III et al., 2020 G3). If sulfides were segregated during magma evolution, it is expected that the residual magmas have higher $d_{66}\text{Zn}$.

3. If sulfide is also a residual phase during partial melting, it is also expected that HIMU lavas have higher $d_{66}\text{Zn}$. The authors used a La/Sm vs. $d_{66}\text{Zn}$ plot to argue against $d_{66}\text{Zn}$ variation being controlled by degree of partial melting. However, La/Sm is not sensitive to sulfide.

4. Mineral-melt isotope fractionation factors used in modeling are not justified. The authors simply refer to two published papers without any proper justification for their selection. The cpx-melt fractionation factor, -0.17 at 1000 K , seems to come from Sossi et al. (2018 Chem Geol). This value was proposed for partial melting of spinel peridotite, which occurs under low pressure. Is it appropriate to use this -0.17 value directly for partial melting under higher pressure when garnet is a residual phase? It is well known that the reduced partition coefficient of mineral increases with increasing pressure. The pressure effect should be evaluated. What about the effect of melt composition on the mineral-melt isotope fractionation factor? The authors prefer a CO_2 -rich source. As a consequence, the melt would also be CO_2 -rich. Does the CO_2 -rich melt composition affect the mineral-melt isotope fractionation factor?

5. The authors prefer a recycled carbonate component for high $d_{66}\text{Zn}$ in HIMU lavas. If so, it is expected that $d_{66}\text{Zn}$ is correlated with geochemical parameters that are controlled by carbonates. Following this logic, I expect to see a correlation between $d_{66}\text{Zn}$ and $^{206}\text{Pb}/^{204}\text{Pb}$. However, this correlation does not exist: Cook-Austral lavas have $^{206}\text{Pb}/^{204}\text{Pb}$ ranging from 19 to 22, but homogeneous $d_{66}\text{Zn}$. The authors observed a negative correlation between SiO_2 and $d_{66}\text{Zn}$, and they argued that this reflects the varying role of carbonate in their mantle source, because increasing CO_2 leads to lower SiO_2 in the melt. However, SiO_2 in lavas is affected by: (a) crystal fractionation; (b) pressure (depth) and degree of partial melting; (c) source lithology, for example, peridotite vs. pyroxenite; (d) volatiles. The authors ignored the first three effects, and jumped directly on the last one. This approach weakens their argument.

Finally, based on Figure 1, high $d_{66}\text{Zn}$ is not a distinctive characteristic of marine carbonates. The majority of carbonates, but NOT ALL, have higher $d_{66}\text{Zn}$ than mantle value.

In summary, the authors present some important and interesting Zn isotope data on

HIMU lavas, which help the community to better constrain the origin of this enigmatic mantle endmember. The data should be published with an improved discussion on the origin of the observed high $\delta^{66}\text{Zn}$.

Response to reviewers' comments:

A complete list of all the concerns and comments of the reviewers, and the corresponding response (in blue text), are given below. We point to contents of response letter by referring to figure numbers marked by the letter “R” (e.g. Figure R1). In some cases, we refer associated revisions of the revised manuscript through line numbers (e.g. line 10-12) or figure numbers (e.g. Figure 2). **The line numbers refer to the clean manuscript.**

**Reviewer #1**

This paper presents data analyses for the isotopic compositions of Zn of HIMU basalts
from Cook-Austral and St. Helena islands. The results presented support previous
suggestions that the mantle source of HIMU involves subducted carbonate via
carbonatite metasomatism (e.g. Weiss et al., 2016). It is important that the new data will
be published as they strengthen our understanding of the evolution of the extreme
HIMU mantle component. However, I believe the authors should better explain and
demonstrate in the text what is the major and unique contribution of the new data. For
example, the authors mention that ‘it is not known how far such carbonate can be
transported into the deeper mantle.’ (Line 34-35) and ‘therefore, demonstrate that
carbonates can be transported into Earth’s deep mantle and recycled back to...’ (Line
52-53); but data collected on the carbon isotopic composition and inclusions of
diamonds originating from the transition zone and lower mantle, indicate the depth to
which crustal carbonate is recycled back to the mantle. The model presented in the
manuscript explains the Zn isotopic composition of the HIMU source, but does not
show or sufficiently discuss it in the context of the other characteristic isotopic
compositions of this mantle component (i.e. Sr, Nd, etc. isotopic compositions). In
addition, the authors provide a few ‘outstanding’ sentences in the text like ‘Earth’s
carbon cycle has played a fundamental role in creating a habitable planet over
geological time.’ (Line 54), ‘our findings for Earth’s deep carbon cycle’ (line 304) and
‘Figure 4 Cartoon depicting Earth’s deep carbon cycle and...’, which sound a bit out
of the scope of the current manuscript.

In summary, I believe major revisions are needed before this manuscript is considered
for publication in Nature Communications.

**Response:** Thank you for the very helpful comments and suggestions! Now we have
revised the introduction section substantially in order to better present the two major
contribution of this study: (1) The heavy Zn isotopic signature of HIMU OIBs provides
convincing isotopic evidence for recycled surficial carbonates in the deep mantle.

Although superdeep diamonds identify ‘crustal carbon’ recycled back to the deep
mantle, their low- $\delta^{13}\text{C}$ values (mean $\delta^{13}\text{C} = -28.3\text{‰}$ to -4.8‰ , mostly $< -20\text{‰}$)
indicate the contribution from subducted organic carbon ($\delta^{13}\text{C}_{\text{organic carbon}} \approx -57\text{‰}$ to
32 -28‰) rather than sedimentary carbonates ($\delta^{13}\text{C}_{\text{carbonates}} \approx 0\text{‰}$) (e.g., *Walter et al., 2011*;
*Cartigny, 2005*; *Thomson et al., 2014*). (2) This study definitely links the famous HIMU
component in the deep mantle to the isotopically heavy carbonates that were formed on
Earth’s surface.

In addition, modelling for Sr and Nd isotopes have been added in the revised
manuscript. The calculated result shows that both Sr and Nd isotopes (6%–7% of
carbonates are required) can coincide with the model given by Zn isotopes (6%–12%
of carbonates are required).

Following your suggestions, those ‘outstanding’ sentences have been deleted or
reworded.

**Detailed comments**

Line 61-62: ‘estimates ranging from minor to major’ – these amount estimations do not
say much; numbers as tonne per year should be given

**Response:** The amount estimations are now mentioned in the revised manuscript.
Please see lines **46-47 in the clean manuscript**. Thank you for your constructive advice.

Line 62-63: ‘it is uncertain whether carbonates in subducted slabs can be transported
into the deep mantle, i.e., into the mantle transition zone and the lower mantle’ –
diamonds from the deep mantle holds direct evidence for the depth to which recycled
carbon is subducted (e.g. *Cartigny, Elements, 2005*; *Thomson et al., CMP, 2014*; *Smith*
*et al., nature, 2018*; *Li et al., EPSL, 2019*), so this statement should be revised or
explained.

**Response:** Superdeep diamonds with the origin of the lower mantle have lighter carbon
isotopic compositions (mean $\delta^{13}\text{C} = -4.8\text{‰}$ to -28.3‰ , mostly $< -20\text{‰}$) than those of

mantle ($\delta^{13}\text{C} \approx -5\%$) (Thomson et al. 2014; Walter et al. 2011; Cartigny, 2005). Such
light carbon isotopic compositions of superdeep diamonds were thought to inherit from
recycled organic carbon ($\delta^{13}\text{C} \approx -57\%$ to -28%) instead of sedimentary carbonates
($\delta^{13}\text{C} \approx 0\%$) (Eigenbrode and Freeman, 2006; Alt and Teagl, 2003; Wilson et al., 2006;
Cartigny, 2005; Walter et al., 2011). Therefore, diamonds from the deep mantle hold
direct evidence for ‘recycled carbon’ rather than ‘recycled sedimentary carbonates’ in
the deep mantle. In our study, the heavy Zn isotopic signature of HIMU lavas is proved
to be definite isotopic evidence for recycled sedimentary carbonates in the deep mantle.
Considering the reviewer’s suggestion, we have revised the whole paragraph. Please
see lines **45-62 in the clean manuscript**.

**References cited in this response:**

- Thomson, A.R., Kohn, S.C., Bulanova, G.P., Smith, C.B., Araujo, D., Walter, M.J., Eimf, 2014. Origin
of sub-lithospheric diamonds from the Juina-5 kimberlite (Brazil): constraints from carbon isotopes
and inclusion compositions. Contributions to Mineralogy and Petrology 168, 1081.
Walter, M.J., Kohn, S.C., Araujo, D., Bulanova, G.P., Smith, C.B., Gaillou, E., Wang, J., Steele, A., Shirey,
S.B., 2011. Deep Mantle Cycling of Oceanic Crust: Evidence from Diamonds and Their Mineral
Inclusions. Science 334, 54-57.
Cartigny, P., 2005. Stable Isotopes and the Origin of Diamond. Elements 1, 79-84.
Eigenbrode, J.L., Freeman, K.H., 2006. Late Archean rise of aerobic microbial ecosystems. Proceedings
of the National Academy of Sciences 103, 15759.
Alt, J.C., Teagle, D.A.H., 2003. Hydrothermal alteration of upper oceanic crust formed at a fast-spreading
ridge: mineral, chemical, and isotopic evidence from ODP Site 801. Chemical Geology 201, 191-
211.
Wilson, D.A.H., Acton, D.S., Coggon, R.M., Teagle, D.A.H., Green, D.R.H., 2006. Data report:
Compositions of calcium carbonate veins from superfast spreading rate crust, ODP Leg 206.
Proceedings of the Ocean Drilling Program Scientific Results 206.

Line 90-94: an additional and important supporting evidence for the connection
between the HIMU basalts and a carbonated mantle source is the similarity in trace
elements between these magmas and deep high-Mg carbonatite encapsulated in
diamonds (Weiss et al., nature, 2016).

**Response:** Great point! We have added this essential evidence in the revised paper.
Please see lines **71-72 in the clean manuscript**.

Line 94-96: this statement is kind of odd regarding the Rohrbach et al. study that is
cited, as in this study the mantle domain that experiences carbonatite redox melting
within the transition zone is formed at first by subduction

**Response:** Thank you for pointing this out. The introduction has now been reworded.

Line 97-98: 'this says little or nothing about the origin of such carbon(ate).' – what
about the other evidence that relates the HIMU source with subduction (e.g. Sulfur
isotopes in HIMU basalts), don't they link the carbonate to subduction?

**Response:** Overall, sulfur, He, isotopes and Os isotopes of the HIMU basalts only
suggest ancient crustal materials were recycled into the HIMU source (*Cabral et al.,*
*2013; Hanyu and Kaneoka, 1997; Roy-Barman and Allègre, 1995*), but they could not
further identify the carbonates components. On the other hand, olivine phenocryst and
melt inclusion provide evidence for deep carbon stored in the HIMU source (*Saal et al.,*
*1998; Weiss et al., 2016*), but it is not as powerful as isotopic evidence in tracing the
origin of such deep carbon. The Zn isotopic evidence in our study, however, definitely
links such deep carbon to recycled surficial carbonates. Nevertheless, we recognize the
reviewer's concern. In the revised manuscript, we have reworded this sentence. Please
see lines **81-83 in the clean manuscript.**

**References cited in this response:**

Cabral, R.A., Jackson, M.G., Rose-Koga, E.F., Koga, K.T., Whitehouse, M.J., Antonelli, M.A., Farquhar,
111 J., Day, J.M.D., Hauri, E.H., 2013. Anomalous sulfur isotopes in plume lavas reveal deep mantle
storage of Archaean crust. *Nature* 496, 490-493.

Hanyu, T., Kaneoka, I., 1997. The uniform and low $^3\text{He}/^4\text{He}$ ratios of HIMU basalts as evidence for their
origin as recycled materials. *Nature* 390, 273-276.

Roy-Barman, M., Allègre, C.J., 1995. $^{187}\text{Os}/^{186}\text{Os}$ in oceanic island basalts: tracing oceanic crust
recycling in the mantle. *Earth and Planetary Science Letters* 129, 145-161.

Saal, A.E., Hart, S.R., Shimizu, N., Hauri, E.H., Layne, G.D., 1998. Pb Isotopic Variability in Melt
Inclusions from Oceanic Island Basalts, Polynesia. *Science* 282, 1481-1484.

Weiss, Y., Class, C., Goldstein, S.L., Hanyu, T., 2016. Key new pieces of the HIMU puzzle from olivines
and diamond inclusions. *Nature* 537, 666-670.

Line 107: for comparison with what?

**Response:** We have re-written the whole sentence. Please see lines **92-95 in the clean**
**manuscript**. Thank you for your advice.

Line 113-114: 'does not necessarily invalidate previously proposed models for the
genesis of HIMU,' – not only that the new observation does not invalidate previously
proposed models, rather, I kind of see that the new observation strengthen/support them.

**Response:** Yes, we agree.

Line 142-143: but few lines above the authors argue that fractional crystallization is not
a major factor in the isotopic signature of Zn (lines 137-140); therefore, can this high
value be explained differently?

**Response:** No, the sentence was still meant to discuss the effect of fractional
crystallization on St. Helena samples. The discussion might be not clear enough. We,
therefore, re-written the discussion about the effect of fractional crystallization on Zn
isotopes of HIMU samples. Please see lines **117-134 in the clean manuscript**.
Furthermore, we modified **Figure 2a** to exclude low-MgO samples (MgO<8wt.%)
because crystal fractionation has the possibility to affect their isotopes.

We still argue that fractional crystallization is not a major factor affecting the Zn
isotopic signature of high-MgO St. Helena samples, based on evidence from natural
samples and quantitative models. (1) The most primitive sample (SH-35 sample,
MgO=15.72wt.%) still displays an elevated $\delta^{66}\text{Zn}$ values (0.36‰), confirming that the
primary magma of St. Helena lavas has the high- $\delta^{66}\text{Zn}$ signature. (2) Modeling results
suggest that even if the $\delta^{66}\text{Zn}$ variation of St. Helena samples is consistent with the
trend affected by fractionation of mafic minerals, the theoretical initial $\delta^{66}\text{Zn}$ value of
St. Helena primary magma should be as high as 0.37‰ (Fig R1).

Figure R1. Modeled variations of $\delta^{66}\text{Zn}$ values during crystal fractionation.

Line 158-171 and 279-300: Mg isotopic signature – The role of these parts in the paper seems slightly weak and unnecessary.

Response: We don't agree. Magnesium stable isotopes of basalts have also been proposed to be a powerful tool for identifying contributions of recycled carbonate-bearing sediments and constraining the nature of mantle source (Huang *et al.*, 2015; Li *et al.*, 2017; Wang *et al.*, 2018; Zeng *et al.*, 2021). A combination of Mg and Zn isotopes of HIMU basalts indicates that although carbonatite metasomatism can modify the Zn isotopes of the mantle, but cannot modify the Mg isotopes of the mantle. Therefore, compared to Mg isotopes, Zn isotopes are more powerful to trace such kind of recycled carbonate in the deep mantle. We have strengthened the discussion of Mg isotopes in lines 298-315 in the clean manuscript of the revised paper.

References cited in this response:

Huang, J., Li, S.-G., Xiao, Y., Ke, S., Li, W.-Y., Tian, Y., 2015. Origin of low $\delta^{26}\text{Mg}$ Cenozoic basalts from South China Block and their geodynamic implications. *Geochimica et Cosmochimica Acta* 164, 298-317.

Li, S.-G., Yang, W., Ke, S., Meng, X., Tian, H., Xu, L., He, Y., Huang, J., Wang, X.-C., Xia, Q., Sun, W., Yang, X., Ren, Z.-Y., Wei, H., Liu, Y., Meng, F., Yan, J., 2017. Deep carbon cycles constrained by a large-scale mantle Mg isotope anomaly in eastern China. *National Science Review* 4, 111-120.

Wang, X.-J., Chen, L.-H., Hofmann, A.W., Hanyu, T., Kawabata, H., Zhong, Y., Xie, L.-W., Shi, J.-H., Miyazaki, T., Hirahara, Y., Takahashi, T., Senda, R., Chang, Q., Vaglarov, B.S., Kimura, J.-I., 2018a.

Recycled ancient ghost carbonate in the Pitcairn mantle plume. Proceedings of the National
Academy of Sciences 115, 8682-8687.

Zeng, G., Chen, L.-H., Hofmann, A.W., Wang, X.-J., Liu, J.-Q., Yu, X., Xie, L.-W., 2021. Nephelinites
in eastern China originating from the mantle transition zone. Chemical Geology 576, 120276.

Line 204: in addition, the strong resemblance in trace elements pattern between HIMU
basalts and high-Mg carbonatite encapsulated in diamonds, indicates the involvement
of carbonatite metasomatism.

**Response:** Good idea! We have added this in lines **198-201 in the clean manuscript**.

Line 253-257: ‘Our modeling (Supplementary information, Part 7) suggests that 6–12%
of originally subducted carbonates are required to generate a HIMU source with $\delta^{66}\text{Zn}$
of 0.30%. Melting of such carbonated peridotite can therefore explain the high- $\delta^{66}\text{Zn}$
signature of the HIMU basalts.’ – how this modeling coincides with the characteristic
HIMU isotopic signature of Sr, Nd, etc.?

**Response:** Good point! We now calculate for Sr and Nd isotopes, and include a brief
discussion that our modeling can coincide with the characteristic HIMU isotopic
signature of the radiogenic isotopes. Please see lines **248-261 in the clean manuscript**,
**Figure 3b** and Figure R2.

Our model for Sr and Nd isotopes suggests that 6%–7% of subducted carbonates
are required to produce the HIMU source. This proportion of subducted carbonates
(6%–7%) can coincide with the estimate given by Zn isotopes (6%–12%) in our model.
About 21%–25% total carbonatite melt can produce the carbonated peridotite with
HIMU isotopic signature of Sr, Nd and Zn isotopes (Figure 3b). Such proportion of
carbonatite melt is consistent with modeling results by radiogenic Os isotopes (18%–
30%) calculated by *Weiss et al. (2016)* and Pb isotopes (25%) calculated by *Chauvel,*
*et al. (1992)*.

**Figure R2. Model for $\delta^{66}\text{Zn}$ versus $^{87}\text{Sr}/^{86}\text{Sr}$ and $^{143}\text{Nd}/^{144}\text{Nd}$.**

**References cited in this response:**

Weiss, Y., Class, C., Goldstein, S.L., Hanyu, T., 2016. Key new pieces of the HIMU puzzle from olivines
 and diamond inclusions. *Nature* 537, 666-670.

Chauvel C, et al. HIMU-EM: The French Polynesian connection. *Earth and Planetary Science Letters*.
 110, 99-119 (1992).

Line269-273: I was always puzzled how a metasomatized carbonated peridotite source
 can survive for long periods in the convective mantle (especially in its deeper parts), as
 such a source is susceptible to melting at ambient mantle temperatures. It will be good
 if the authors can provide some insight on that subject.

**Response:** We speculate that even if the high temperature could theoretically lead to
 melting of such metasomatized source, extremely high pressure (>22 GPa) might have
 the inverse effect. The high pressure would limit the expansion of volume which is led
 by the melting of metasomatized materials and also restrain the migration of such melt.
 In addition, the appropriate phase changes can enhance the stability of the
 metasomatized carbonated peridotite source (*Bina and Helffrich, 2014*). Distinct
 density between metasomatized carbonated peridotite and lower mantle prevents
 metasomatized materials from being easily modified by ambient mantle, leading that
 such chemical heterogeneity survived in the lower mantle (*Bina and Helffrich, 2014*).
 Furthermore, it is possible that some part of the deep mantle close to the core-mantle
 boundary has not been involved in the mantle convections, otherwise, no endmembers,
 EM1, EM2, and HIMU can survive in the mantle.

**References cited in this response:**

Bina, C.R., Helffrich, G., 2014. 3.2 - Geophysical Constraints on Mantle Composition, in: Holland, H.D.,
Turekian, K.K. (Eds.), Treatise on Geochemistry (Second Edition). Elsevier, Oxford, pp. 41-65.

Line 285-288: I am not convinced that high Ti/Eu ratios and HIMU-like isotopic
signature in xenoliths, by themselves, are strong enough evidence for carbonatitic
metasomatism impact. Do the Zealandia xenoliths show any other geochemical features
that support such a suggestion?

**Response:** *Scott et al. (2014)* provide the following evidence to argue that Zealandia
xenoliths experienced carbonatite metasomatism: (1) Some of Zealandia xenoliths
belong to wehrlites and olivine-rich lherzolites. Besides, apatite and hydrous minerals
can be observed in these Zealandia xenoliths. (2) Clinopyroxene of Zealandia xenoliths
are enriched in LREE and display negative anomalies for Nb, Ta, Zr, Hf and Ti. (3)
Clinopyroxene of Zealandia xenoliths have low Ti/Eu ratios and high Th/U ratios (up
to ~7) as a result of carbonatite metasomatism (*Scott et al., 2014; Wittig et al., 2010*).
We have improved the description of Zealandia xenoliths based on the study of *Scott et*
*al. (2014)*. Please see lines **299-304 in the clean manuscript**.

**Figures of Scott et al. (2014):** (A), (B) and (C) are averaged clinopyroxene trace element patterns
for North Otago peridotites normalized to C1 Chondrite.; (D), (E) and (F) are averaged
clinopyroxene trace element patterns for East Otago peridotites normalized to C1 Chondrite.

**References cited in this response:**

Scott, J.M., Waight, T.E., van der Meer, Q.H.A., Palin, J.M., Cooper, A.F., Münker, C., 2014.
Metasomatized ancient lithospheric mantle beneath the young Zealandia microcontinent and its role
in HIMU-like intraplate magmatism. *Geochemistry, Geophysics, Geosystems* 15, 3477-3501.

Wittig, N., Pearson, D.G., Duggen, S., Baker, J.A., Hoernle, K., 2010. Tracing the metasomatic and
magmatic evolution of continental mantle roots with Sr, Nd, Hf and Pb isotopes: A case study
of Middle Atlas (Morocco) peridotite xenoliths. *Geochimica et Cosmochimica Acta* 74, 1417-1435.

Line 289-290: reference?

**Response:** Revised. Please see lines **304-306 in the clean manuscript**. Thank you!

Line 303: ‘summarizes the implication of our findings for the Earth’s deep carbon cycle’
– this statement is highly exaggerating the impact of new data on our understanding of
the carbon cycle.

**Response:** Revised. Please see lines **321-322 in the clean manuscript**. Thank you for
your advice.

Line 312-315: in the model of Weiss et al. (2016) the high-U/Pb signature is related to
the subducting slab (which is part of the model).

**Response:** Thank you for your response. We have reworded this sentence. Please see
lines **330-335 in the clean manuscript**.

Line 319: here again, I am puzzled by how such a metasomatic source with a solidus
below the ambient mantle can be preserved for long periods (i.e. billions of years) at
the convective mantle.

**Response:** Please see above comment. Thank you for your response.

Line 342-349: sounds like this last part is over the scope of the present manuscript.

**Response:** Reworded. We have simplified this part and made the content more closely
related to the present manuscript. Thank you! Please see lines **361-363 in the clean**

**manuscript.**

Figure 1: the line is between 0.1 and 0.2, and not as stated in the captions.

**Response:** Revised. Please see caption of Figure 1. Thank you.

Figure 2: in (a), it would be good to show where the recycled carbonate is situated with
respect to the general trend observed for $\delta^{66}\text{Zn}$ vs. SiO_2 .

**Response:** Revised. Please see Figure 2. Thank you for your advice.

Figure 3: Is the carbonatite metasomatism arrow direction based on a specific model?
or determined based on the position of carbonatitic melts in the $\delta^{66}\text{Zn}$ vs. $^{187}\text{Os}/^{188}\text{Os}$
space? if not, based on what it was determined?

**Response:** The arrow direction is based on the position of carbonatitic melts. Figure 3
has been revised following the above comment. Thank you for your advice.

Figure 4: Although many subduction and plume figures have been drawn over the years,
this specific one shares too many graphic features and details with the one published
by Weiss et al. (2016). Perhaps the authors should consider altering this Figure a bit? or
mentioning in the captions something like 'altered after Weiss et al., 2016'?

**Response:** Revised. Now the new figure is quite different from the one of Weiss et al.
(2016).

Sincerely,

Yaakov Weiss

The Fredy and Nadine Herrmann Institute of Earth Sciences

The Hebrew University, Jerusalem 91904, Israel

Reviewer #2:

Review of “Zinc isotopic evidence for recycled carbonate in the deep mantle” by Zhang
et al.

Zhang et al. report high precision Zn isotope data for HIMU lavas from St. Helena and
Cook-Austral. They found that HIMU lavas have higher $\delta^{66}\text{Zn}$ than other OIBs,
MORB, and peridotites, which they attributed to a signature of recycled carbonate in
the HIMU mantle sources. The data are important and should be published. The
interpretation can be improved.

My major concern is whether high $\delta^{66}\text{Zn}$ a distinctive SOURCE signature of HIMU.
The authors did discuss and evaluate the effects of crystal fractionation and partial
melting, and concluded that their effects too small to explain the observed high $\delta^{66}\text{Zn}$
in HIMU lavas. However, the discussion is not complete.

**Response:** Thank you for the suggestions! Now we have strengthened the discussion
for the effects of crystal fractionation and partial melting.

Figure S2 shows all published data together with new data from this study. It is obvious
that some published non-HIMU OIBs (Kilauea Iki lavas from Hawaii and Hekla lavas
from Iceland from Chen et al., 2013) have $\delta^{66}\text{Zn}$ as high as HIMU lavas. High $\delta^{66}\text{Zn}$
in some low MgO Kilauea Iki and Hekla lavas may be a result of extensive crystal
fractionation. Hence it is important to group them together to show the effect of crystal
fractionation. Crystal fractionation effect is also observed in St Helena lavas, but not
obvious in Cook-Austral lavas. – Why?

**Response:** We agree that those samples from Hawaii and Iceland have experienced
extensive crystal fractionation and their Zn isotopic compositions have been affected
by such a process. However, the crystal fractionation effect is not obvious for Cook-
Austral lavas in this study because most of them have high MgO contents ($\text{MgO} > 8\text{wt.}\%$,
please see Figure R3) and only experience a small amount of fractionation of mafic
minerals (e.g. ol, cpx). The small amount of fractionation of mafic minerals would not

lead to significant variation of Zn isotopic compositions (Chen et al., 2013; McCoy-
West et al., 2018; Wang et al., 2017).

To avoid misunderstanding of Figure 2a, the two samples with low MgO contents
(MgO<8wt.%) of HIMU basalts have been removed from **Figure 2a**, and for the cited
MORBs and other OIBs, only those MgO>8 wt.% samples are shown on Figure 2a.
Thus, the unevolved samples (MgO>8wt.%) in Figure 2a are applicable to discuss the
nature of the mantle source. Meanwhile, all published data together with new data from
this study are presented on Supplementary Figure 1 to show the distinct correlations
generated by source difference and mineral fractionation, respectively.

**Figure R3. MgO contents versus SiO₂ contents.**

**References cited in this response:**

Chen, H., Savage, P.S., Teng, F.-Z., Helz, R.T., Moynier, F., 2013. Zinc isotope fractionation during
magmatic differentiation and the isotopic composition of the bulk Earth. *Earth and Planetary
Science Letters* 369-370, 34-42.

McCoy-West, A.J., Fitton, J.G., Pons, M.-L., Inglis, E.C., Williams, H.M., 2018. The Fe and Zn isotope
composition of deep mantle source regions: Insights from Baffin Island picrites. *Geochimica et
Cosmochimica Acta* 238, 542-562.

Wang, Z.-Z., Liu, S.-A., Liu, J., Huang, J., Xiao, Y., Chu, Z.-Y., Zhao, X.-M., Tang, L., 2017. Zinc isotope
fractionation during mantle melting and constraints on the Zn isotope composition of Earth's upper
mantle. *Geochimica et Cosmochimica Acta* 198, 151-167.

Role of sulfide during crystal fractionation. Sulfide inclusions in olivines have been
reported in HIMU lavas (Cabral et al., 2013 Nature; Delavault et al., 2016 PNAS;
Dottin III et al., 2020 G3). If sulfides were segregated during magma evolution, it is
expected that the residual magmas have higher $\delta^{66}\text{Zn}$.

**Response:** Thank you for pointing it out. We argue that the effect of sulfide segregation
can be ruled out by the following three points.

(1) Although OIBs from Reunion Island had experienced extensive sulfide segregation
(Peters et al., 2016; Peters et al., 2019), their $\delta^{66}\text{Zn}$ values are indistinguishable from
those of MORBs ($\sim 0.28\text{‰}$ in average) (Wang et al., 2017, see figure below). Thus,
sulfide segregation has limited effect on Zn isotopes and cannot explain high- $\delta^{66}\text{Zn}$
signature of HIMU samples.

**Figure in Wang et al., 2017.**

(2) Quantitative calculations also indicate that even if we assumed an extremely and
unreasonably high $\alpha^{\text{sulfide-cpx}}$ with value of 0.995 (assumed $\delta^{66}\text{Zn}$ value of sulfide
could be as low as -5‰), zinc isotopes of melt would not be significantly elevated
($<0.05\text{‰}$) during sulfide segregation (Figure R4a). Therefore, the high- $\delta^{66}\text{Zn}$ signature
of HIMU samples cannot result from sulfide segregation.

(3) As Ni is extremely compatible in both sulfide and olivine (Li et al., 2003;
Chowdhury et al., 2021; Adam and Green, 2006), we modelled the variation of Ni
contents in olivine during sulfide segregation (Figure R4b). The calculated result shows

that Ni contents of olivine decrease sharply if sulfide segregation exists. However, such
 a significant variation of Ni contents is inconsistent with that of olivine from HIMU
 OIBs. Instead, nickel contents in olivine from HIMU OIBs are consistent with the
 calculated olivine liquid line without sulfide (the gray dashed line in Figure R4b,
 *Herzberg et al., 2014*), as inferred by *Weiss et al. (2016)*. Therefore, HIMU lavas haven't
 experienced obvious sulfide segregation.

**Figure R4. (a) $\delta^{66}\text{Zn}$ versus Ni contents of whole-rock from HIMU OIBs.** The red line reflects
 the melt from normal peridotite. **(b) Ni contents versus Fo values of olivine from HIMU OIBs.**
 This figure is altered from *Weiss et al. (2016)* and *Herzberg et al. (2014)*. All data are also from *Weiss*
 *et al. (2016)*. The gray dashed line is the calculated olivine liquid line of descent (*Herzberg et al.,*
 *2014*). The green line denotes the variation of Ni contents in olivine caused by sulfide segregation.

**References cited in this response:**

Weiss, Y., Class, C., Goldstein, S.L., Hanyu, T., 2016. Key new pieces of the HIMU puzzle from olivines
 and diamond inclusions. *Nature* 537, 666-670.
 Adam, J., Green, T., 2006. Trace element partitioning between mica- and amphibole-bearing garnet
 lherzolite and hydrous basanitic melt: 1. Experimental results and the investigation of controls on
 partitioning behaviour. *Contributions to Mineralogy and Petrology* 152, 1-17.
 Li, C., Ripley, E.M., Naldrett, A.J., 2003. Compositional Variations of Olivine and Sulfur Isotopes in the
 Noril'sk and Talnakh Intrusions, Siberia: Implications for Ore-Forming Processes in Dynamic
 Magma Conduits. *Economic Geology* 98, 69-86.
 Chowdhury, P., Dasgupta, R., Phelps, P.R., Lee, C.-T.A., Anselm, R.A., 2021. Partitioning of chalcophile
 and highly siderophile elements (HSEs) between sulfide and carbonated melts – Implications for
 HSE systematics of kimberlites, carbonatites, and melt metasomatized mantle domains. *Geochimica*
 *et Cosmochimica Acta* 305, 130-147.

Herzberg, C., Cabral, R.A., Jackson, M.G., Vidito, C., Day, J.M.D., Hauri, E.H., 2014. Phantom Archean
crust in Mangaia hotspot lavas and the meaning of heterogeneous mantle. *Earth and Planetary*
*Science Letters* 396, 97-106.

Peters, B.J., Day, J.M.D., Taylor, L.A., 2016. Early mantle heterogeneities in the Réunion hotspot source
inferred from highly siderophile elements in cumulate xenoliths. *Earth and Planetary Science*
*Letters* 448, 150-160.

Peters, B.J., Shahar, A., Carlson, R.W., Day, J.M.D., Mock, T.D., 2019. A sulfide perspective on iron
isotope fractionation during ocean island basalt petrogenesis. *Geochimica et Cosmochimica Acta*
245, 59-78.

Wang, Z.-Z., Liu, S.-A., Liu, J., Huang, J., Xiao, Y., Chu, Z.-Y., Zhao, X.-M., Tang, L., 2017. Zinc isotope
fractionation during mantle melting and constraints on the Zn isotope composition of Earth's upper
mantle. *Geochimica et Cosmochimica Acta* 198, 151-167.

If sulfide is also a residual phase during partial melting, it is also expected that HIMU
lavas have higher $\delta^{66}\text{Zn}$. The authors used a La/Sm vs. $\delta^{66}\text{Zn}$ plot to argue against
$\delta^{66}\text{Zn}$ variation being controlled by the degree of partial melting. However, La/Sm is
not sensitive to sulfide.

**Response:** Thank you for pointing it out. We recognize the reviewer's concern and now
use Os contents to estimate the effect of sulfide, because Os is the most sensitive
element to sulfide (*Bockrath et al., 2004; Chowdhury et al., 2021*). No correlation
($R^2=0.035$, $p=0.63$) can be found between Os contents and $\delta^{66}\text{Zn}$ values (Figure R5a),
suggesting that the high- $\delta^{66}\text{Zn}$ signature of HIMU samples is not caused by the effect
of sulfide. What's more, quantitative calculations also indicate that the effect of sulfide
during partial melting on Zn isotopes of HIMU lavas is negligible (Figure R5b), due to
a low proportion (0.00025%) of sulfide in the mantle (*Chowdhury et al., 2021*).

**Figure R5. $\delta^{66}\text{Zn}$ versus Os contents of HIMU OIBs.**

**References cited in this response:**

Bockrath, C., Ballhaus, C., Holzheid, A., 2004. Fractionation of the Platinum-Group Elements During
Mantle Melting. *Science* 305, 1951-1953.

Chowdhury, P., Dasgupta, R., Phelps, P.R., Lee, C.-T.A., Anselm, R.A., 2021. Partitioning of chalcophile
and highly siderophile elements (HSEs) between sulfide and carbonated melts – Implications for
HSE systematics of kimberlites, carbonatites, and melt metasomatized mantle domains. *Geochimica
et Cosmochimica Acta* 305, 130-147.

Mineral-melt isotope fractionation factors used in modeling are not justified. The
authors simply refer to two published papers without any proper justification for their
selection. The cpx-melt fractionation factor, -0.17 at 1000 K, seems to come from Sossi
et al. (2018 *Chem Geol*). This value was proposed for partial melting of spinel peridotite,
which occurs under low pressure. Is it appropriate to use these -0.17 values directly for
partial melting under higher pressure when garnet is a residual phase? It is well known
that the reduced partition coefficient of mineral increases with increasing pressure. The
pressure effect should be evaluated.

**Response:** Thanks for your suggestions. The new discussion about this issue has been
added in the revised manuscript. Please see lines **146-151 in the clean manuscript**. We
also discussed whether the selected mineral-melt isotope fractionation factors in the
quantitative modeling are reasonable in the supplementary materials. Please see lines
**378-385** in supplementary information.

Previous studies have suggested that the degree of Zn isotope fractionation

induced by melting in the garnet stability field model is very similar to that estimated
for melting in the spinel stability field (*McCoy-West et al., 2018; Sossi et al., 2018*).
Observations on natural samples also support this proposal. The OIBs produced by
melting of CO₂-free garnet peridotitic mantle have indistinguishable Zn isotopic
compositions from MORB that produced by melting of spinel peridotite (Figure R6a).
Thus, melting of garnet- and spinel-facies mantle has similar effect on Zn isotopic
fractionation (*McCoy-West et al., 2018; Sossi et al., 2018; Wang et al., 2017*), and it is
reasonable to use fractionation factor given by *Sossi et al. (2018)*. This is also supported
by the following observations which argue against any obvious pressure effect on Zn
isotopic fractionation:

(1) Non-metasomatized peridotites in *Wang et al. (2017)* span a large range of both
melting degrees (5–40%) and initial pressures of melting (1–5 GPa), but they display
homogenous $\delta^{66}\text{Zn}$ values ($0.18 \pm 0.06\text{‰}$). Thus, *Wang et al. (2017)* argued that melting
pressures should not be the major factor controlling the Zn isotope fractionation during
mantle melting.

(2) The $\alpha_{\text{grt-cpx}}$ in Figure R6b is estimated from equilibrium inter-mineral Zn isotope
fractionation between Grt and Cpx of eclogites measured by *Huang et al. (2022)*. There
is no variation in $\alpha_{\text{grt-cpx}}$ with increasing pressure (from 3GPa to 7GPa, *Chen et al.,*
*2020*), indicating that pressure has negligible effect on Zinc isotopic fractionation.

(3) We can compare HIMU lavas with other OIBs, because all of these samples are
from similar depths under high pressure where garnet is the residue phase. The pressure
effect, if exists, would have a similar effect on both HIMU lavas and the other OIBs.
However, HIMU lavas show higher $\delta^{66}\text{Zn}$ values relative to the other OIBs (Figure R7).
Therefore, we argue that pressure effect cannot explain the high- $\delta^{66}\text{Zn}$ values in HIMU
lavas.

**Figure R6. (a) $\delta^{66}\text{Zn}$ versus SiO_2 contents of HIMU OIBs. (b) $\alpha_{\text{grt-cpx}}$ versus pressure of Grt and**
 **Cpx in eclogite.**

**References cited in this response:**

McCoy-West, A.J., Fitton, J.G., Pons, M.-L., Inglis, E.C., Williams, H.M., 2018. The Fe and Zn isotope
 composition of deep mantle source regions: Insights from Baffin Island picrites. *Geochimica et*
 *Cosmochimica Acta* 238, 542-562.

Sossi, P.A., Nebel, O., O'Neill, H.S.C., Moynier, F., 2018. Zinc isotope composition of the Earth and its
 behaviour during planetary accretion. *Chemical Geology* 477, 73-84.

Wang, Z.-Z., Liu, S.-A., Liu, J., Huang, J., Xiao, Y., Chu, Z.-Y., Zhao, X.-M., Tang, L., 2017. Zinc isotope
 fractionation during mantle melting and constraints on the Zn isotope composition of Earth's upper
 mantle. *Geochimica et Cosmochimica Acta* 198, 151-167.

Huang, J., Huang, J.-X., Griffin, W.L., Huang, F., 2022. Zn-, Mg- and O-isotope evidence for the origin
 of mantle eclogites from Roberts Victor kimberlite (Kaarvaal Craton, South Africa). *Geology*.

Chen, C., et al. 2020. Compositional and pressure controls on calcium and magnesium isotope
 fractionation in magmatic systems. *Geochimica et Cosmochimica Acta*. 290, 257-270.

What about the effect of melt composition on the mineral-melt isotope fractionation
 factor? The authors prefer a CO_2 -rich source. As a consequence, the melt would also
 be CO_2 -rich. Does the CO_2 -rich melt composition affect the mineral-melt isotope
 fractionation factor?

**Response:** The effect of CO_2 on the fractionation factor is still unknown. Nevertheless,
 if CO_2 could affect the melt composition, the increase of CO_2 concentration of melt
 leads to the decrease of non-bridging oxygen (NBO) of melt (*Morizet et al., 2014*). The
 decreasing NBO would lead to the increase of coordination of metals (*Farges et al.,*
 *2004; Jackson et al., 2005; Sossi et al., 2018*). As Zn with higher coordination and

weaker Zn-O bonds is preferentially partitioned with the lighter isotopes (*Schauble,*
*2004*), it might provide a possibility for the enrichment of light Zn isotopes in CO₂-rich
melt rather than heavy Zn isotopes.

**References cited in this response:**

- Morizet, Y., Paris, M., Gaillard, F., Scaillet, B., 2014. Carbon dioxide in silica-undersaturated melt Part
II: Effect of CO₂ on quenched glass structure. *Geochimica et Cosmochimica Acta* 144, 202-216.
Farges, F., Lefrère, Y., Rossano, S., Berthereau, A., Calas, G., Brown, G.E., 2004. The effect of redox
state on the local structural environment of iron in silicate glasses: a combined XAFS spectroscopy,
molecular dynamics, and bond valence study. *Journal of Non-Crystalline Solids* 344, 176-188.
Jackson, W.E., Farges, F., Yeager, M., Mabrouk, P.A., Rossano, S., Waychunas, G.A., Solomon, E.I.,
Brown, G.E., 2005. Multi-spectroscopic study of Fe(II) in silicate glasses: Implications for the
coordination environment of Fe(II) in silicate melts. *Geochimica et Cosmochimica Acta* 69, 4315-
4332.
Sossi, P.A., Nebel, O., O'Neill, H.S.C., Moynier, F., 2018. Zinc isotope composition of the Earth and its
behaviour during planetary accretion. *Chemical Geology* 477, 73-84.
Schauble, E.A., 2004. Applying Stable Isotope Fractionation Theory to New Systems. *Reviews in*
*Mineralogy and Geochemistry* 55, 65-111.

The authors prefer a recycled carbonate component for high d₆₆Zn in HIMU lavas. If
so, it is expected that d₆₆Zn is correlated with geochemical parameters that are
controlled by carbonates. Following this logic, I expect to see a correlation between
d₆₆Zn and ²⁰⁶Pb/²⁰⁴Pb. However, this correlation does not exist: Cook-Austral lavas
have ²⁰⁶Pb/²⁰⁴Pb ranging from 19 to 22, but homogeneous d₆₆Zn.

**Response:** Thank you for your advice. Our modelling reveals that a certain degree of
melts from DMM mix with melts from pure HIMU components results in variations of
²⁰⁶Pb/²⁰⁴Pb values (Figure R7). For Zn isotopes, depleted mantle (DMM) have lower
Zn contents than melts from HIMU source. A low proportion (<50%) of melt from
DMM has less effect on δ⁶⁶Zn values of primary magma from the HIMU source.
Therefore, most of Cook-Austral lavas display HIMU-like δ⁶⁶Zn values.

**Figure R7. $\delta^{66}\text{Zn}$ versus $^{206}\text{Pb}/^{204}\text{Pb}$.**

The authors observed a negative correlation between SiO_2 and $\delta^{66}\text{Zn}$, and they argued
 that this reflects the varying role of carbonate in their mantle source, because increasing
 CO_2 leads to lower SiO_2 in the melt. However, SiO_2 in lavas is affected by: (a) crystal
 fractionation; (b) pressure (depth) and degree of partial melting; (c) source lithology,
 for example, peridotite vs. pyroxenite; (d) volatiles. The authors ignored the first three
 effects and jumped directly on the last one. This approach weakens their argument.

**Response:** Thank you for pointing it out. A new discussion about the variation of SiO_2
 contents is now included in the revised manuscript. Please see lines 263-270 in the
 **clean manuscript.**

(a) For crystal fractionation: All samples (e.g. MORB, HIMU lavas, other OIBs and
 continental nephelinites) display no correlations between MgO and SiO_2 contents
 (Figure R8). Furthermore, even though St. Helena samples underwent crystal
 fractionation, which gives rise to a large variation of MgO contents (5.5–15.7wt.%), the
 SiO_2 of these samples only vary from 43.5 wt.% to 46.9wt.% (Figure R6a). Therefore,
 the effect by crystal fractionation on their SiO_2 contents is negligible.

**Figure R8. Major element variations for samples with MgO > 8 wt.%.**

(b) For pressure and degree of melting: Experimental melts shows that even if the
 pressure and the melt percent vary from 3 GPa to 7 GPa and from 0 to 100%
 respectively, SiO₂ contents of partial melts only can change in the range of 45–48 wt.%
 (Figure R9), indicating that pressure has less effect on the SiO₂ contents of partial melts
 (Walter, 1998).

**Figure R9. Variation diagram showing SiO₂ abundance vs melt percent (%). This figure is**
 **modified from Figure 5 of Walter (1998).**

(c) For source lithology: We summarize the SiO₂ contents of experimental melts (Figure
 R10). Although different source lithology can produce the melt with variable SiO₂
 contents, SiO₂-unsaturated melt (SiO₂ < 43 wt.%, e.g. HIMU lavas and continental

nephelinites) cannot release from the CO₂-free source (e.g. peridotite, silica-
 deficient/excess pyroxenite, 43–66 wt.%). Overall, we think that source lithology has
 limited effect on variation of SiO₂ contents under CO₂-free condition.

 **Figure R10. Variation diagram showing SiO₂ contents.** Experimental melts of peridotite are from
 *Dasgupta et al. (2007), Dasgupta et al. (2013), Hirose(1997), Davis et al.(2011), Hirose and*
 *Kushiro (1993), Takahashi (1986) and Walter (1998)*. Experimental melts of pyroxenite are from
 *Dasgupta et al.(2006), Gerbode and Dasgupta (2010), Kiseeva et al.(2012), Pertermann and*
 *Hirschmann (2003), Spandler et al. (2008), Yasuda et al. (1994), Yaxley and Green (1998), Yaxley*
 *and Sobolev (2007), Hirschmann et al. (2003), Keshav et al. (2004), Kogiso and Hirschmann (2006)*
 *and Kogiso et al. (2003)*.

**References cited in this response:**

Hirose, K., Kushiro, I., 1993. Partial melting of dry peridotites at high pressures: Determination of
 compositions of melts segregated from peridotite using aggregates of diamond. *Earth and Planetary*
 *Science Letters* 114, 477-489.

Walter, M.J., 1998. Melting of Garnet Peridotite and the Origin of Komatiite and Depleted Lithosphere.
 *Journal of Petrology* 39, 29-60.

Dasgupta, R., Hirschmann, M.M., Smith, N.D., 2007. Partial Melting Experiments of Peridotite + CO₂
 at 3 GPa and Genesis of Alkalic Ocean Island Basalts. *Journal of Petrology* 48, 2093-2124.

Dasgupta, R., Mallik, A., Tsuno, K., Withers, A.C., Hirth, G., Hirschmann, M.M., 2013. Carbon-dioxide-
 rich silicate melt in the Earth's upper mantle. *Nature* 493, 211-215.

Hirose, K., 1997. Melting experiments on Iherzolite KLB-1 under hydrous conditions and generation of
 high-magnesian andesitic melts. *Geology* 25, 42-44.

Davis, F.A., Hirschmann, M.M., Humayun, M., 2011. The composition of the incipient partial melt of
 garnet peridotite at 3GPa and the origin of OIB. *Earth and Planetary Science Letters* 308, 380-390.

Takahashi, E., 1986. Melting of a dry peridotite KLB-1 up to 14 GPa: Implications on the Origin of
 peridotitic upper mantle. *Journal of Geophysical Research: Solid Earth* 91, 9367-9382.

Dasgupta, R., Hirschmann, M.M., Stalker, K., 2006. Immiscible Transition from Carbonate-rich to
Silicate-rich Melts in the 3 GPa Melting Interval of Eclogite + CO₂ and Genesis of Silica-
undersaturated Ocean Island Lavas. *Journal of Petrology* 47, 647-671.

Gerbode, C., Dasgupta, R., 2010. Carbonate-fluxed Melting of MORB-like Pyroxenite at 2.9 GPa and
Genesis of HIMU Ocean Island Basalts. *Journal of Petrology* 51, 2067-2088.

Kiseeva, E.S., Yaxley, G.M., Hermann, J., Litasov, K.D., Rosenthal, A., Kamenetsky, V.S., 2012. An
Experimental Study of Carbonated Eclogite at 3 center dot 5-5 center dot 5 GPa-Implications for
Silicate and Carbonate Metasomatism in the Cratonic Mantle. *Journal of Petrology* 53, 727-759.

Pertermann, M., Hirschmann, M.M., 2003. Anhydrous Partial Melting Experiments on MORB-like
Eclogite: Phase Relations, Phase Compositions and Mineral–Melt Partitioning of Major Elements
at 2–3 GPa. *Journal of Petrology* 44, 2173-2201.

Spandler, C., Yaxley, G., Green, D.H., Rosenthal, A., 2008. Phase relations and melting of anhydrous k-
bearing eclogite from 1200 to 1600 degrees C and 3 to 5 GPa. *Journal of Petrology* 49, 771-795.

Yasuda, A., Fujii, T., Kurita, K., 1994. Melting phase relations of an anhydrous mid-ocean ridge basalt
from 3 to 20 GPa: Implications for the behavior of subducted oceanic crust in the mantle. *Journal*
*of Geophysical Research: Solid Earth* 99, 9401-9414.

Yaxley, G.M., Green, D.H., 1998. Reactions between eclogite and peridotite: Mantle refertilisation by
subduction of oceanic crust. *Schweizerische Mineralogische Und Petrographische Mitteilungen* 78,
243-255.

Yaxley, G.M., Sobolev, A.V., 2007. High-pressure partial melting of gabbro and its role in the Hawaiian
magma source. *Contributions to Mineralogy and Petrology* 154, 371-383.

Hirschmann, M.M., Kogiso, T., Baker, M.B., Stolper, E.M., 2003. Alkalic magmas generated by partial
melting of garnet pyroxenite. *Geology* 31, 481-484.

Keshav, S., Gudfinnsson, G.H., Sen, G., Fei, Y., 2004. High-pressure melting experiments on garnet
clinopyroxenite and the alkalic to tholeiitic transition in ocean-island basalts. *Earth and Planetary*
*Science Letters* 223, 365-379.

Kogiso, T., Hirschmann, M.M., 2006. Partial melting experiments of bimineralic eclogite and the role of
recycled mafic oceanic crust in the genesis of ocean island basalts. *Earth and Planetary Science*
*Letters* 249, 188-199.

Kogiso, T., Hirschmann, M.M., Frost, D.J., 2003. High-pressure partial melting of garnet pyroxenite:
possible mafic lithologies in the source of ocean island basalts. *Earth and Planetary Science Letters*
216, 603-617.

Finally, based on Figure 1, high d₆₆Zn is not a distinctive characteristic of marine
carbonates. The majority of carbonates, but NOT ALL, have higher d₆₆Zn than mantle
value.

**Response:** We agree. Low-δ⁶⁶Zn values of carbonates may be reset by hydrothermal

fluids or post-depositional diagenesis (*Liu et al., 2022; Lv et al., 2018*). Some biogenic
carbonates could also display low- $\delta^{66}\text{Zn}$ values (*Zhao et al., 2021*). However, the
majority of carbonates show high $\delta^{66}\text{Zn}$ values.

**References cited in this response:**

Liu, S.-A., Qu, Y.-R., Wang, Z.-Z., Li, M.-L., Yang, C., Li, S.-G., 2022. The fate of subducting carbon
tracked by Mg and Zn isotopes: A review and new perspectives. *Earth-Science Reviews* 228, 104010.

Lv, Y., Liu, S.-A., Wu, H., Hohl, S.V., Chen, S., Li, S., 2018. Zn-Sr isotope records of the Ediacaran
Doushantuo Formation in South China: diagenesis assessment and implications. *Geochimica et*
*Cosmochimica Acta* 239, 330-345.

Zhao, M., Tarhan, L.G., Zhang, Y., Hood, A., Asael, D., Reid, R.P., Planavsky, N.J., 2021. Evaluation of
shallow-water carbonates as a seawater zinc isotope archive. *Earth and Planetary Science Letters*
553, 116599.

In summary, the authors present some important and interesting Zn isotope data on
HIMU lavas, which help the community to better constrain the origin of this enigmatic
mantle endmember. The data should be published with an improved discussion on the
origin of the observed high $\delta^{66}\text{Zn}$.

**Response:** We appreciate your constructive and helpful comments/suggestions. We
have carefully revised the manuscript in light of all the comments, and we hope to have
rigorously addressed the shortcomings of our original manuscript.

Reviewer #1 (Remarks to Author) :

Second review

Manuscript Number: NCOMMS-21-23139A

Title: Zinc isotopic evidence for recycled carbonate in the deep mantle

Authors: Xiao-Yu Zhang, Li-Hui Chen, Xiao-Jun Wang, Takeshi Hanyu, Albrecht W. Hofmann, Tsuyoshi Komiya, Kentaro Nakamura, Yasuhiro Kato, Gang Zeng, Wen-Xian Gou, Wei-Qiang Li

Overall, the authors addressed my original concerns and comments regarding this research and its results. Although I am not yet convinced regarding some of the arguments they provided, it is not justified for me to prevent publication. Nonetheless, I do find a place to point up some issues that I believe are important/significant:

- 1) The low $\delta^{13}\text{C}$ signature in diamonds indicates the recycling of surface carbon to the deep mantle, both of organic and sedimentary origin. The organic signature is just easier to distinguish compared to the sedimentary which is not a lot different from that of the mantle. Therefore, I suggest rewording lines 51-60.
- 2) Line 60-62: I think there is no place for the second part of the sentence - ‘....., and this hinders our full understanding of the deep carbon cycle.’
- 3) Line 77: reference 17 is not suitable for the point dealing with olivine composition.
- 4) Line 81-83: I do not agree with this sentence. Overall the geochemistry of HIMU basalts indicates the involvement of recycled components, and it is more than likely that the carbonate component is of the same origin.
- 5) I still think the Mg isotope composition part is weak and irrelevant/redundant - sections: ‘Normal $\delta^{26}\text{Mg}$ signature of the HIMU Mantle Sources’ and ‘Mg isotopic puzzle of HIMU’.
- 6) Most of the section ‘Origin of the HIMU Component’ is just a repetition of part of the introduction (lines 168-201; some of it is perhaps even irrelevant - lines 179-185).

- 7) I am not sure I understand from the text how continental nephelinites are interwoven into the story? in other words, their relevance is not clearly explained?
- 8) Line 333-335: I suggest the authors look at studies that deal with the trace element signature of deep metasomatic agents (high-density fluids - HDF) that are trapped in diamonds (e.g. Tomlinson et al., EPSL, 2009; Weiss et al., EPSL, 2011, 2013). The negative anomalies of alkalis are a fingerprint of such fluids and when they metasomatize peridotite, they imprint this signature which in turn is transferred to other mantle melts upon melting of the metasomatized peridotite (e.g. alkali basalts/basanites).

Sincerely,

Yaakov Weiss

The Fredy and Nadine Herrmann Institute of Earth Sciences

The Hebrew University, Jerusalem 91904, Israel

Email: yakov.weiss@mail.huji.ac.il

Reviewer #3 (Remarks to the Author):

The manuscript of Zhang et al provide data of Zn isotopes for HIMU as well as complete set of whole-rock major and trace element and Os-Sr isotopes. They propose that the anomalously high $d66Zn$ values indicate deep recycling of sedimentary carbonates in the HIMU source. Overall, I think this observation is excellent and should be published as it shed light on the genesis of the important HIMU mantle endmember. I read through the authors responses to the former reviewers and find most of problems are addressed, but I list two problems raised by the former reviewer #2 that have not been well addressed. I also list my own major questions below, which I hope the authors will carefully consider. Anyway, I believe after careful revision they will make the manuscript a great scientific contribution.

Problems of responses to former Reviewers:

(1) Their response to Reviewer #2 on possible correlation of $d66Zn$ vs. $206Pb/204Pb$ is not complete. They have not provided the calculation detail for the figure. In fact, DMM melts do not have very low Zn compared to OIB as Zn is a moderately incompatible element, in contrast, a HIUM melt seems to have higher Pb contents, thus, an opposite curve is expected?

(2) Problem of use of SiO_2 . I agree with Reviewer #2 that their plot of SiO_2 vs. $d66Zn$ is not significant, and the authors' responses in fact further prove that SiO_2 tends to be affected by a number of factors. They want $d66Zn$ to measure the effect of HIMU, but very low SiO_2 is not unique for HIMU OIBs. Many OIBs have very low SiO_2 , but their $d66Zn$ are normal. What they need to do is plot $d66Zn$ with isotopes/elements that characterize the HIMU mantle, e.g., U/Pb, $206/204$ et al.

Problems to be considered:

(1) The current manuscript is unclear on how carbonates with heavy Zn isotopes can be transported to deep mantle through plate subduction. In fact, it is generally considered that subducted carbonates (mainly in form of $CaCO_3$) is unstable in the subduction zone and would react with mantle rocks, resulting in transition of $CaCO_3$ to Mg-rich carbonates. They should discuss how heavy Zn is inherited by such new Mg-rich carbonates. Moreover, carbonate melts, if subducted plate melts, would be released and react with overlying mantle wedge (Thomson et al., 2016-Nature), thus, what is the fate of Zn after in these processes? I suggest they discuss more about these processes, then, this will help readers understand how such heavy Zn can survive the subduction zone process and enter the deep mantle.

(2) Regarding their explanation of normal mantle-like Mg isotopes, it looks that such Mg isotopes are not good tracer for recycling of subducted marine carbonates. In other words, carbonate is unnecessary to explain the normal mantle-like Mg-isotopes, right? As stated by themselves, there are several many studies showing that subducted carbonates have indeed contributed to light Mg isotope in other continental and oceanic OIBs (Huang et al., 2015; Li et al., 2017; Wang et al., 2018; Zeng et al., 2021), but why not for HIMU here? Maybe I suggest the authors to carefully consider and discuss the genetic differences between the HIMU samples and other continental and oceanic OIBs (light in Mg isotopes). The comparison maybe helpful for understanding how the HIMU signatures are produced and their genetic difference from other light-Mg isotope OIBs. If not I tend to agree with the former Reviewer #1, and suggest them to remove the discussion on Mg isotopes.

(3) Interpretation of the effect of CO_2 on olivine Ca and Mn/Fe ratio. In fact, olivines precipitated from carbonated melts unnecessarily have high Ca contents, because Ca tends to participate in carbonated melts during olivine crystallization (see the papers of Dasgupta). I also suggest the authors to read the paper of Gavrilenko et al (2016). The authors need to either delete this part or make some modifications according to Gavrilenko et al (2016). Similarly, if they compile the data in the papers of Mallik and Dasgupta, 2012 & 2013; Matzen et al., 2017, they would find Mn is similar to Ca in carbonated melts and has a lower content in olivine. Thus, the high Mn/Fe and Ca in

olivine cannot be directly used to indicate a carbonated melt. This part must be properly addressed before publication!

(4) Their statement about the genesis of HIMU signature in the mantle. While I agree that HIMU needs a pre-existing high U/Pb ratio in the mantle, a high U/Pb ratio is not simply derived from seafloor alteration. In fact, seafloor alteration increases both U and Pb significantly, a reason why subduction processes can produce a HIMU component is that Pb is more easily extracted by subduction fluids/melts, thus, a high U/Pb tends to be maintained in the subducted oceanic crust. The authors need to make this clearer here. Moreover, a high Th/Pb would not result in high $^{206}\text{Pb}/^{204}\text{Pb}$, there is a mistake here.

(5) The authors are incorrect by stating that seafloor alteration would result in elevated Th and thus high Th/Pb. Seafloor alteration generally would not result in elevated Th. A high Th/Pb ratio in the subducted oceanic crust is also resulted mainly from the preferential loss of Pb during slab-dehydration and melting. This part should be significantly revised in the next submission.

(6) A relevant paper (Zinc isotope constraints on carbonated mantle sources for rejuvenated-stage lavas from Kaua'i, Hawai'i. <https://doi.org/10.1016/j.chemgeo.2022.120967>) should be cited somewhere in this manuscript.

Lines 172-175: The authors suggest seafloor alteration caused the final enrichment of U and Th in the ROC, which is incorrect. In fact, MORBs are generally more enriched in U-Th than depleted mantle, thus, this is the reason for the time-integrated $^3\text{He}/^4\text{He}$ in such a mantle component.

Lines 177-179: At any time authors want show olivine Ni and Mn/Fe are high or not, they should indicate olivine Fo. In fact, only Fo-rich olivines are useful to indicate mantle source lithology! Moreover, in Lines 74-77, the authors state that high olivine Mn/Fe ratios indicate the role of CO₂, but why they show that the Mn/Fe ratios here directly point to peridotitic source? Something is unreasonable here.

Lines 182-185: The authors should make it clearer why Herzburg's model is not applicable here by showing some more details. It is not good to hide such important information in the Supplementary files.

Lines 195-198: I disagree with the direct use of olivine Ca and Ca/Al ratio, which are both sensitive to CO₂ content and lithology in the mantle. They should either delete these statement or explore more on how CO₂ influences olivine Ca and Ca/Al.

Line 198-201: I don't find this sentence informative. This sentence should be modified to show what specific characteristics point to carbonated melts.

Lines 279-285: The authors should explain here how such heavy Zn can survive the subduction processes, since CaCO₃ would not be stable in the subduction zone, and it would either melt or be transformed to other carbonate phases.

**Response to reviewers' comments:**

A complete list of all the concerns and comments of the reviewers, and the corresponding response
(in blue text), are given below. Figures in this response are marked in the form of Figure R+Number
(e.g. Figure R1), which are different from those in the revised manuscript (e.g. Figure 1) and in the
supplementary materials (e.g. Figure S1). **The line numbers refer to the tracked manuscript.**

**Reviewer #1 (Remarks to the Author):**

Overall, the authors addressed my original concerns and comments regarding this research and its
results. Although I am not yet convinced regarding some of the arguments they provided, it is not
justified for me to prevent publication. Nonetheless, I do find a place to point up some issues that I
believe are important/significant:

1) The low $\delta^{13}\text{C}$ signature in diamonds indicates the recycling of surface carbon to the deep mantle,
both of organic and sedimentary origin. The organic signature is just easier to distinguish compared
to the sedimentary which is not a lot different from that of the mantle. Therefore, I suggest rewording
lines 51-60.

**Response:** Thank you very much! We agree your interpretations about the carbon isotopic
compositions of diamonds. This part has been revised according to your suggestion (see lines **51-**
**63**):

“Although studies on superdeep diamonds and their high-pressure mineral inclusions (e.g.,
$\text{Ca}(\text{Ti}, \text{Si})\text{O}_3$ -perovskite and calcic-majoritic garnet) have suggested the deep cycling of surficial
carbon into the lower mantle⁶, the low- $\delta^{13}\text{C}$ signatures (mean $\delta^{13}\text{C} = -28.3\%$ to -4.8% ^{7,8}, mostly $<$
21 -20%) of those diamonds are mainly inherited from subducted organic carbon ($\delta^{13}\text{C}_{\text{organic carbon}} \approx -57\%$
to -28% ⁹) rather than sedimentary carbonates ($\delta^{13}\text{C} \approx 0\%$ ^{10,11}). Recently reported high $\delta^{18}\text{O}$ values
($+9.1\%$ to $+10.5\%$) of some high-pressure mineral inclusions in superdeep diamonds, together with
the host diamonds’ carbon and nitrogen isotopic compositions, were regarded as evidence for
subduction of carbonated igneous oceanic crust, rather than sedimentary carbonates, into transition-
zone depths¹². Therefore, convincing isotopic evidence is still lacking to distinguish the signature
of sedimentary carbonates that transported into the deep mantle.”

2) Line 60-62: I think there is no place for the second part of the sentence - ‘....., and this hinders
our full understanding of the deep carbon cycle.’

**Response:** We agree. Deleted.

3) Line 77: reference 17 is not suitable for the point dealing with olivine composition.

**Response:** We agree. Deleted.

4) Line 81-83: I do not agree with this sentence. Overall the geochemistry of HIMU basalts indicates
the involvement of recycled components, and it is more than likely that the carbonate component is
of the same origin.

**Response:** We agree. This sentence has been revised. Please see lines **83-86**:

“Thus far, convincing isotopic tracer for recycled carbonates can further strengthen the genetic
link between the carbonate component in the HIMU mantle sources and the recycled surficial
carbonates.”

5) I still think the Mg isotope composition part is weak and irrelevant/redundant - sections: ‘Normal
$\delta^{26}\text{Mg}$ signature of the HIMU Mantle Sources’ and ‘Mg isotopic puzzle of HIMU’.

**Response:** Thank you! You’ve successfully persuaded us to rethink the reason to keep Mg isotopes
in this paper. Because we haven’t provided enough justification in the introduction section that Mg
isotopes are also useful to trace the carbonated source of HIMU basalts, which makes the Mg isotope
parts weak and irrelevant/redundant. Therefore, we decide to delete all descriptions of Mg isotopes,
although it is not easy to make such a decision...

6) Most of the section ‘Origin of the HIMU Component’ is just a repetition of part of the introduction
(lines 168-201; some of it is perhaps even irrelevant – lines 179-185).

**Response:** We agree. This section has been shortened significantly. Please see lines **197-216**.

7) I am not sure I understand from the text how continental nephelinites are interwoven into the
story? in other words, their relevance is not clearly explained?

**Response:** One sentence is added to explain why these continental nephelinites are needed in the
story (see lines **296-299**):

“The continental nephelinites from eastern China, which have been regarded as direct melts
derived from recycled carbonate-bearing crustal materials⁵², display the highest $\delta^{66}\text{Zn}$ values
(0.40‰–0.50‰)²³.”

8) Line 333-335: I suggest the authors look at studies that deal with the trace element signature of
deep metasomatic agents (high-density fluids - HDF) that are trapped in diamonds (e.g. Tomlinson
et al., EPSL, 2009; Weiss et al., EPSL, 2011, 2013). The negative anomalies of alkalis are a
fingerprint of such fluids and when they metasomatize peridotite, they imprint this signature which
in turn is transferred to other mantle melts upon melting of the metasomatized peridotite (e.g. alkali
basalts/basanites).

**Response:** Good idea! We agree that the negative anomalies of alkalis e.g., Rb, K, are a fingerprint
of such fluids (Tomlinson et al., 2009; Weiss et al., 2011; Weiss et al., 2013; Weiss et al., 2015), and
when they metasomatize peridotite, they imprint this signature which in turn is transferred to other

mantle melts upon melting of the metasomatized peridotite (e.g. alkali basalts/basanites). However,
we still need a mechanism to generate such a kind of elemental signature. We note that apart from
the depletion of K and Rb, HIMU lavas also display quite low Ba/Th ratios (with a mean of 60.8,
<http://georoc.mpch-mainz.gwdg.de/georoc>), which are distinct from the high Ba/Th ratios of the
HDFs (High-Density Fluids) (with a mean of 169.9, *Weiss et al., 2015*). Therefore, trace elemental
characteristics of HIMU basalts are not exactly the same as HDFs. The extremely low Ba/Th can be
attributed to the effect of K-hollandite (named recently as liebermannite, *Ma et al., 2018*) formation
in the mantle transition zone (*Mazza et al., 2019; Zeng et al., 2021*). Potassium (K), barium (Ba)
and rubidium (Rb) are retained in the residual K-hollandite, while Th is largely extracted with the
melts, resulting in extremely low K, Rb and Ba/Th ratios in the released carbonate melts. Mantle
peridotite infiltrated by such melts will inherit the low Ba/Th ratios of the carbonate liquids.
Nevertheless, we don't deny the other possible mechanisms to generate such a kind of elemental
signature of HIMU basalts.

**References cited in this response:**

- Tomlinson, E.L., Müller, W., Eimf, 2009. A snapshot of mantle metasomatism: Trace element analysis of coexisting
fluid (LA-ICP-MS) and silicate (SIMS) inclusions in fibrous diamonds. *Earth. Planet. Sci. Lett.* 279, 362-372.
Weiss, Y., Griffin, W.L., Bell, D.R., Navon, O., 2011. High-Mg carbonatitic melts in diamonds, kimberlites and the
sub-continental lithosphere. *Earth. Planet. Sci. Lett.* 309, 337-347.
Weiss, Y., Griffin, W.L., Navon, O., 2013. Diamond-forming fluids in fibrous diamonds: The trace-element
perspective. *Earth. Planet. Sci. Lett.* 376, 110-125.
Weiss, Y., McNeill, J., Pearson, D.G., Nowell, G.M., Ottley, C.J., 2015. Highly saline fluids from a subducting slab
as the source for fluid-rich diamonds. *Nature* 524, 339-342.
88 Ma, C., Tschauer, O., Beckett, J.R., Rossman, G.R., Prescher, C., Prakapenka, V.B., Bechtel, H.A., MacDowell, A.,
2018. Liebermannite, KAlSi₃O₈, a new shock-metamorphic, high-pressure mineral from the Zagami Martian
meteorite. *Meteor. Planet. Sci.* 53, 50-61.
Mazza, S.E., Gazel, E., Bizimis, M., Moucha, R., Béguélin, P., Johnson, E.A., McAleer, R.J., Sobolev, A.V., 2019.
Sampling the volatile-rich transition zone beneath Bermuda. *Nature* 569, 398-403.
Zeng, G., Chen, L.-H., Hofmann, A.W., Wang, X.-J., Liu, J.-Q., Yu, X., Xie, L.-W., 2021. Nephelinites in eastern
China originating from the mantle transition zone. *Chem. Geol.* 576, 120276.

Sincerely,

Yaakov Weiss

The Fredy and Nadine Herrmann Institute of Earth Sciences

The Hebrew University, Jerusalem 91904, Israel

Email: yakov.weiss@mail.huji.ac.il

**Reviewer #3 (Remarks to the Author):**

The manuscript of Zhang et al provide data of Zn isotopes for HIMU as well as complete set of
whole-rock major and trace element and Os-Sr isotopes. They propose that the anomalously high
$d_{66}\text{Zn}$ values indicate deep recycling of sedimentary carbonates in the HIMU source. Overall, I
think this observation is excellent and should be published as it shed light on the genesis of the
important HIMU mantle endmember. I read through the authors responses to the former reviewers
and find most of problems are addressed, but I list two problems raised by the former reviewer #2
that have not been well addressed. I also list my own major questions below, which I hope the
authors will carefully consider. Anyway, I believe after careful revision they will make the
manuscript a great scientific contribution.

**Problems of responses to former Reviewers:**

(1) Their response to Reviewer #2 on possible correlation of $d_{66}\text{Zn}$ vs. $^{206}\text{Pb}/^{204}\text{Pb}$ is not complete.
They have not provided the calculation detail for the figure. In fact, DMM melts do not have very
low Zn compared to OIB as Zn is a moderately incompatible element, in contrast, a HIUM melt
seems to have higher Pb contents, thus, an opposite curve is expected?

**Response:** Thank you for your comments!

The calculated curve in Figure R1 is used to explain the two samples from Anatonu-stage of
Raivavae Island which show less radiogenic Pb isotopes ($^{206}\text{Pb}/^{204}\text{Pb}=19.3$ and 19.7) but have
similar Zn isotopic composition compared to those classic HIMU OIBs. Previous Sr, Nd, Pb isotopic
study suggests that these Anatonu-stage samples might be produced by mixing of a small proportion
of HIMU melt and the Anatonu DM-type melt (Miyazaki *et al.*, 2018). Such Anatonu DM-type melt
is intrinsically heterogeneous and it might be generated by mixing between the melt from DM-type
plume source and that from a kind of enriched component, e.g. recycled seamount (alkalic basalts)
(Miyazaki *et al.*, 2018). Considering the complicated petrogenesis of Anatonu-stage samples, we
adopt their two-stage model (Miyazaki *et al.*, 2018) and optimize the modelling.

Firstly, the melt from DM-type plume source mixed with the melt from a kind of enriched
component (recycled seamount?) and produced the Anatonu DM-type melt. Here, the $^{206}\text{Pb}/^{204}\text{Pb}$
ratio, $\delta^{66}\text{Zn}$ value, Zn and Pb contents of the DM-type melt are assumed as 18.28 (averaged by
Global MORB samples, <http://georoc.mpch-mainz.gwdg.de/georoc>), 0.25‰ (calculated in
Supplementary Tables), 55ppm (McDonough and Sun, 1995) and 0.61ppm (Gale *et al.*, 2013),

respectively. The $^{206}\text{Pb}/^{204}\text{Pb}$ ratio, $\delta^{66}\text{Zn}$ value, Zn and Pb contents of the melt from the enriched
component are assumed as 19.11 (Miyazaki et al., 2018), 0.33‰ (averaged by all of OIBs), 100ppm
and 4.5ppm, respectively.

Secondly, the Anatonu DM-type melt mixed with the HIMU melt and generated the Anatonu-
stage lava. The calculated Pb and Zn content of Anatonu DM-type melt are 3.3ppm and 91ppm,
respectively. The calculated $^{206}\text{Pb}/^{204}\text{Pb}$ ratio and $\delta^{66}\text{Zn}$ value of Anatonu DM-type melt are assumed
as 19.08 and 0.33‰, respectively. We use the geochemical composition of one Cook-Austral sample
(RRT302-1), which displays the highest $^{206}\text{Pb}/^{204}\text{Pb}$ values (21.66) to represent the HIMU melt. Its
Pb, Zn content and $\delta^{66}\text{Zn}$ value are 1.4ppm, 91.3ppm and 0.38‰, respectively. Finally, mixing of
Anatonu DM-type melt and a small proportion of HIMU melt might produce two Anatonu-stage
samples with low $^{206}\text{Pb}/^{204}\text{Pb}$ values (Miyazaki et al., 2018).

**Figure R1. $\delta^{66}\text{Zn}$ versus $^{206}\text{Pb}/^{204}\text{Pb}$.**

**References cited in this response:**

Miyazaki, T., Hanyu, T., Kimura, J.-I., Senda, R., Vaglarov, B.S., Chang, Q., Hirahara, Y., Takahashi, T., Kawabata,
H., Sato, T., 2018. Clinopyroxene and bulk rock Sr–Nd–Hf–Pb isotope compositions of Raivavae ocean island
basalts: Does clinopyroxene record early stage magma chamber processes? *Chem. Geol.* 482, 18-31.

McDonough, W.F., Sun, S.s., 1995. The composition of the Earth. *Chem. Geol.* 120, 223-253.

Gale, A., Dalton, C.A., Langmuir, C.H., Su, Y., Schilling, J.-G., 2013. The mean composition of ocean ridge basalts.
*Geochem. Geophys. Geosyst.* 14, 489-518.

(2) Problem of use of SiO₂. I agree with Reviewer #2 that their plot of SiO₂ vs. $\delta^{66}\text{Zn}$ is not
significant, and the authors' responses in fact further prove that SiO₂ tends to be affected by a
number of factors. They want $\delta^{66}\text{Zn}$ to measure the effect of HIMU, but very low SiO₂ is not unique
for HIMU OIBs. Many OIBs have very low SiO₂, but their $\delta^{66}\text{Zn}$ are normal. What they need to
do is plot $\delta^{66}\text{Zn}$ with isotopes/elements that characterize the HIMU mantle, e.g. U/Pb, $^{206}\text{Pb}/^{204}\text{Pb}$ et al.

**Response:** We disagree. We have already addressed this question clearly in the last response. Here
we copy the relevant contents from the last response. Moreover, you may misunderstand the usage
of the plot of SiO₂ vs. Zn isotopes, and here we add some new interpretations to avoid possible
misunderstanding.

Firstly, we have carefully summarized a great number of previous studies to exclude the effect
by these factors. It is apparent that these effects are really limited and cannot lead such apparent
variations of SiO₂ contents (39.2wt.%–50.6wt.%) in Figure 2.

(a) For crystal fractionation: All samples (e.g. MORB, HIMU lavas, other OIBs and continental
nephelinites) display no correlations between MgO and SiO₂ contents (Figure R2). Furthermore,
even though St. Helena samples underwent crystal fractionation, which gives rise to a large variation
of MgO contents (5.5–15.7wt.%), the SiO₂ of these samples only vary from 43.5 wt.% to 46.9wt.%.
Therefore, the effect by crystal fractionation on their SiO₂ contents is negligible.

**Figure R2. Major element variations for samples with MgO > 8wt.%.**

(b) For pressure and degree of melting: Experimental melts shows that even if the pressure and
the melt percent vary from 3GPa to 7GPa and from 0 to 100% respectively, SiO₂ contents of partial
melts can only change in the range of 45–48wt.% (Figure R3), indicating that pressure has less effect
on the SiO₂ contents of partial melts (Walter, 1998).

**Figure R3 Variation diagram showing SiO₂ abundance vs melt percent (%). This figure is modified from**

**Figure 5 of Walter (1998).**

(c) For source lithology: We summarize the SiO₂ contents of experimental melts (Figure R4).
Although different source lithology can produce the melt with variable SiO₂ contents, SiO₂-
unsaturated melt (SiO₂<43wt.%, e.g. HIMU lavas and continental nephelinites) cannot release from
the CO₂-free source (e.g. peridotite, silica-deficient/excess pyroxenite, 43–66wt.%). Overall, we
think that source lithology has limited effect on variation of SiO₂ contents under CO₂-free condition.

**Figure R4. Variation diagram showing SiO₂ contents.** Experimental melts of peridotite are from *Dasgupta et al.*
(2007), *Dasgupta et al. (2013)*, *Hirose (1997)*, *Davis et al. (2011)*, *Hirose and Kushiro (1993)*, *Takahashi (1986)* and
*Walter (1998)*. Experimental melts of pyroxenite are from *Dasgupta et al. (2006)*, *Gerbode and Dasgupta (2010)*,
*Kiseeva et al. (2012)*, *Pertermann and Hirschmann (2003)*, *Spandler et al. (2008)*, *Yasuda et al. (1994)*, *Yaxley and*
*Green (1998)*, *Yaxley and Sobolev (2007)*, *Hirschmann et al. (2003)*, *Keshav et al. (2004)*, *Kogiso and Hirschmann*
(2006) and *Kogiso et al. (2003)*.

Secondly, the plot of $\delta^{66}\text{Zn}$ vs SiO₂ is used to assess the contribution of recycled carbonates,
not the HIMU component, in the mantle source. HIMU lavas are not the only product from mantle
sources with recycled carbonates. It's clear that both continental nephelinites from Eastern China
and Hawaii rejuvenated-stage lavas are not HIMU-type. They originate from sources that were
affected by recycled carbonates and therefore display low SiO₂ contents and high $\delta^{66}\text{Zn}$ values. In
addition, the high $^{206}\text{Pb}/^{204}\text{Pb}$ HIMU reservoir should be generated by long-term radiogenic
accumulation of ^{206}Pb .

Therefore, we still think that SiO₂ content is still suitable in this plot (Fig. 2a).

**References cited in this response:**

Hirose, K., Kushiro, I., 1993. Partial melting of dry peridotites at high pressures: Determination of compositions of
melts segregated from peridotite using aggregates of diamond. *Earth and Planetary Science Letters* 114, 477-

489.

Walter, M.J., 1998. Melting of Garnet Peridotite and the Origin of Komatiite and Depleted Lithosphere. *Journal of*
*Petrology* 39, 29-60.

Dasgupta, R., Hirschmann, M.M., Smith, N.D., 2007. Partial Melting Experiments of Peridotite + CO₂ at 3 GPa and
Genesis of Alkalic Ocean Island Basalts. *Journal of Petrology* 48, 2093-2124.

Dasgupta, R., Mallik, A., Tsuno, K., Withers, A.C., Hirth, G., Hirschmann, M.M., 2013. Carbon-dioxide-rich silicate
melt in the Earth's upper mantle. *Nature* 493, 211-215.

Hirose, K., 1997. Melting experiments on Iherzolite KLB-1 under hydrous conditions and generation of high-
magnesian andesitic melts. *Geology* 25, 42-44.

Davis, F.A., Hirschmann, M.M., Humayun, M., 2011. The composition of the incipient partial melt of garnet
peridotite at 3GPa and the origin of OIB. *Earth and Planetary Science Letters* 308, 380-390.

Takahashi, E., 1986. Melting of a dry peridotite KLB-1 up to 14 GPa: Implications on the Origin of peridotitic upper
mantle. *Journal of Geophysical Research: Solid Earth* 91, 9367-9382.

Dasgupta, R., Hirschmann, M.M., Stalker, K., 2006. Immiscible Transition from Carbonate-rich to Silicate-rich
Melts in the 3 GPa Melting Interval of Eclogite + CO₂ and Genesis of Silica-undersaturated Ocean Island Lavas.
*Journal of Petrology* 47, 647-671.

Gerbode, C., Dasgupta, R., 2010. Carbonate-fluxed Melting of MORB-like Pyroxenite at 2.9 GPa and Genesis of
HIMU Ocean Island Basalts. *Journal of Petrology* 51, 2067-2088.

Kiseeva, E.S., Yaxley, G.M., Hermann, J., Litasov, K.D., Rosenthal, A., Kamenetsky, V.S., 2012. An Experimental
Study of Carbonated Eclogite at 3 center dot 5-5 center dot 5 GPa-Implications for Silicate and Carbonate
Metasomatism in the Cratonic Mantle. *Journal of Petrology* 53, 727-759.

Pertermann, M., Hirschmann, M.M., 2003. Anhydrous Partial Melting Experiments on MORB-like Eclogite: Phase
Relations, Phase Compositions and Mineral-Melt Partitioning of Major Elements at 2-3 GPa. *Journal of*
*Petrology* 44, 2173-2201.

Spandler, C., Yaxley, G., Green, D.H., Rosenthal, A., 2008. Phase relations and melting of anhydrous k-bearing
eclogite from 1200 to 1600 degrees C and 3 to 5 GPa. *Journal of Petrology* 49, 771-795.

Yasuda, A., Fujii, T., Kurita, K., 1994. Melting phase relations of an anhydrous mid-ocean ridge basalt from 3 to 20
226 GPa: Implications for the behavior of subducted oceanic crust in the mantle. *Journal of Geophysical Research:*
*Solid Earth* 99, 9401-9414.

Yaxley, G.M., Green, D.H., 1998. Reactions between eclogite and peridotite: Mantle refertilisation by subduction of
oceanic crust. *Schweizerische Mineralogische Und Petrographische Mitteilungen* 78, 243-255.

Yaxley, G.M., Sobolev, A.V., 2007. High-pressure partial melting of gabbro and its role in the Hawaiian magma
source. *Contributions to Mineralogy and Petrology* 154, 371-383.

Hirschmann, M.M., Kogiso, T., Baker, M.B., Stolper, E.M., 2003. Alkalic magmas generated by partial melting of
garnet pyroxenite. *Geology* 31, 481-484.

Keshav, S., Gudfinnsson, G.H., Sen, G., Fei, Y., 2004. High-pressure melting experiments on garnet clinopyroxenite
and the alkalic to tholeiitic transition in ocean-island basalts. *Earth and Planetary Science Letters* 223, 365-379.

Kogiso, T., Hirschmann, M.M., 2006. Partial melting experiments of bimineralec eclogite and the role of recycled
mafic oceanic crust in the genesis of ocean island basalts. *Earth and Planetary Science Letters* 249, 188-199.

Kogiso, T., Hirschmann, M.M., Frost, D.J., 2003. High-pressure partial melting of garnet pyroxenite: possible mafic
lithologies in the source of ocean island basalts. *Earth and Planetary Science Letters* 216, 603-617.

**Problems to be considered:**

(1) The current manuscript is unclear on how carbonates with heavy Zn isotopes can be transported
to deep mantle through plate subduction. In fact, it is generally considered that subducted carbonates
(mainly in form of CaCO₃) is unstable in the subduction zone and would react with mantle rocks,
resulting in transition of CaCO₃ to Mg-rich carbonates. They should discuss how heavy Zn is
inherited by such new Mg-rich carbonates. Moreover, carbonate melts, if subducted plate melts,
would be released and react with overlying mantle wedge (Thomson et al., 2016-Nature), thus, what
is the fate of Zn after in these processes? I suggest they discuss more about these processes, then,
this will help readers understand how such heavy Zn can survive the subduction zone process and
enter the deep mantle.

**Response:** We agree. Thank you! The process that how heavy Zn survive in the subduction zone
and then enter the deep mantle has been thoroughly discussed recently (*Liu et al., 2022*). Here we
add a brief summary in the revised manuscript (Line 253-257):

“Some kinds of Mg-rich carbonates have significantly high Zn contents (e.g., 147p.p.m in
dolomite and 449p.p.m in magnesite)⁴⁷. Such Mg-rich carbonates are more stable in the subduction
zone^{3,48}, and a considerable proportion of them has the potential to be retained in subducted slabs
and finally introduced into the deep mantle, e.g., the mantle transition zone (410–660km)^{48,49}.”

**References cited in this response:**

Liu, S.-A., Qu, Y.-R., Wang, Z.-Z., Li, M.-L., Yang, C., Li, S.-G., 2022. The fate of subducting carbon tracked by
259 Mg and Zn isotopes: A review and new perspectives. *Earth-Sci. Rev.* 228, 104010.

(2) Regarding their explanation of normal mantle-like Mg isotopes, it looks that such Mg isotopes
are not good tracer for recycling of subducted marine carbonates. In other words, carbonate is
unnecessary to explain the normal mantle-like Mg-isotopes, right? As stated by themselves, there
are several many studies showing that subducted carbonates have indeed contributed to light Mg
isotope in other continental and oceanic OIBs (Huang et al., 2015; Li et al., 2017; Wang et al., 2018;
Zeng et al., 2021), but why not for HIMU here? Maybe I suggest the authors to carefully consider
and discuss the genetic differences between the HIMU samples and other continental and oceanic
OIBs (light in Mg isotopes). The comparison maybe helpful for understanding how the HIMU
signatures are produced and their genetic difference from other light-Mg isotope OIBs. If not I tend

to agree with the former Reviewer #1, and suggest them to remove the discussion on Mg isotopes.

**Response:** According to the reviewer #1's suggestion, we have removed the discussion on Mg
isotopes. Thank you for your comments!

(3) Interpretation of the effect of CO₂ on olivine Ca and Mn/Fe ratio. In fact, olivines precipitated
from carbonated melts unnecessarily have high Ca contents, because Ca tends to participate in
carbonated melts during olivine crystallization (see the papers of Dasgupta). I also suggest the
authors to read the paper of Gavrilenko et al (2016). The authors need to either delete this part or
make some modifications according to Gavrilenko et al (2016). Similarly, if they compile the data
in the papers of Mallik and Dasgupta, 2012 & 2013; Matzen et al., 2017, they would find Mn is
similar to Ca in carbonated melts and has a lower content in olivine. Thus, the high Mn/Fe and Ca
in olivine cannot be directly used to indicate a carbonated melt. This part must be properly addressed
before publication!

**Response:** Thank you for your suggestion! For the effect of CO₂, we agree that Mn/Fe ratio is
useless and unnecessary here. Therefore, we have deleted the description of Mn/Fe ratios in the
introduction (see lines 75-79).

However, we don't agree your interpretation on Ca concentrations of olivine. The olivines in
the paper of Gavrilenko et al. (2016) crystallize from the lavas from Kamchatka Arc and Central
America Arc, and their varied Ca contents are dominated by H₂O contents of arc magma
(Gavrilenko et al., 2016). However, HIMU OIBs are produced by oceanic intraplate volcanism
(Weiss et al., 2016) and their olivines are crystallized from anhydrous magmas. Therefore, it is
unreasonable to make modifications here according to the paper of Gavrilenko et al. (2016).

**Figure from Weiss et al. (2016).**

HIMU olivines are characterized by distinctly high Ca/Al ratios that are far outside the range

previously reported for olivines from MORB and Hawaii lavas (e.g., Loihi and Koolau). The very
high Ca/Al ratio in HIMU olivines compared with Hawaii, Iceland and MORB reflects the distinct
composition of the mantle source of HIMU lavas and points to carbonatite metasomatism (*Weiss et*
*al.*, 2016). We have reworded the sentence in lines **212-213**:

“Such metasomatized peridotite is enriched by carbonatitic fluids with low SiO₂, Al₂O₃ and
high CaO contents^{16,41}.”

**References cited in this response:**

Weiss, Y., Class, C., Goldstein, S.L., Hanyu, T., 2016. Key new pieces of the HIMU puzzle from olivines and
diamond inclusions. *Nature* 537, 666-670.

Gavrilenko, M., Herzberg, C., Vidito, C., Carr, M.J., Tenner, T., Ozerov, A., 2016. A Calcium-in-Olivine
Geothermometer and its Application to Subduction Zone Magmatism. *J. Petrol.* 57, 1811-1832.

(4) Their statement about the genesis of HIMU signature in the mantle. While I agree that HIMU
needs a pre-existing high U/Pb ratio in the mantle, a high U/Pb ratio is not simply derived from
seafloor alteration. In fact, seafloor alteration increases both U and Pb significantly, a reason why
subduction processes can produce a HIMU component is that Pb is more easily extracted by
subduction fluids/melts, thus, a high U/Pb tends to be maintained in the subducted oceanic crust.
The authors need to make this clearer here. Moreover, a high Th/Pb would not result in high
²⁰⁶Pb/²⁰⁴Pb, there is a mistake here.

**Response:** We agree. Revised. Thank you! Please see lines **177-181**:

“Hydrothermal alteration and dehydration processes of subducting crust can induce strong
fractionation of U, Th and Pb, so as to attain the high U/Pb and Th/Pb ratios^{38,39}. Such high U/Pb
and Th/Pb ratios can generate the extremely radiogenic Pb isotopic compositions in HIMU mantle
source with long-term isolation^{38,39}.”

(5) The authors are incorrect by stating that seafloor alteration would result in elevated Th and thus
high Th/Pb. Seafloor alteration generally would not result in elevated Th. A high Th/Pb ratio in the
subducted oceanic crust is also resulted mainly from the preferential loss of Pb during slab-
dehydration and melting. This part should be significantly revised in the next submission.

**Response:** We agree. Revised. Thank you! Please see lines **177-181**.

(6) A relevant paper (Zinc isotope constraints on carbonated mantle sources for rejuvenated-stage
lavas from Kaua'i, Hawai'i. <https://doi.org/10.1016/j.chemgeo.2022.120967>) should be cited
somewhere in this manuscript.

**Response:** We agree. It has been included in the revised manuscript and Figure 2. Thank you! This
is a very useful paper. Please see lines **119-121**.

Lines 172-175: The authors suggest seafloor alteration caused the final enrichment of U and Th in
the ROC, which is incorrect. In fact, MORBs are generally more enriched in U-Th than depleted
mantle, thus, this is the reason for the time-integrated $^3\text{He}/^4\text{He}$ in such a mantle component.

**Response:** Sorry. We haven't mentioned the seafloor alteration in this sentence.

The original sentence:

"The relative enrichment of Th and U in the recycled ancient oceanic crust can also account
for the low- $^3\text{He}/^4\text{He}$ signature of HIMU lavas, because α decay of U and Th will increase ^4He
contents and lower $^3\text{He}/^4\text{He}$ ratios^{34,37,40}."

Lines 177-179: At any time authors want show olivine Ni and Mn/Fe are high or not, they should
indicate olivine Fo. In fact, only Fo-rich olivines are useful to indicate mantle source lithology!
Moreover, in Lines 74-77, the authors state that high olivine Mn/Fe ratios indicate the role of CO₂,
but why they show that the Mn/Fe ratios here directly point to peridotitic source? Something is
unreasonable here.

**Response:** We're sorry for this misunderstanding! This misunderstanding is induced by our
incorrect wording of this sentence. We planned to use high olivine Mn/Fe ratios to suggest a
peridotite source and high olivine Ca contents to suggest a carbonated source, respectively. Then,
we'll have a kind of carbonated peridotite source. Unfortunately, these two lines of evidence are
incorrectly presented and which makes your misunderstanding that high olivine Mn/Fe ratios could
indicate the role of CO₂. We have now deleted this ambiguous sentence. Please see **75-79**.

Lines 182-185: The authors should make it clearer why Herzburg's model is not applicable here by
showing some more details. It is not good to hide such important information in the Supplementary
files.

**Response:** We agree. Revised. Thank you! Please see lines **191-196**:

"However, the calculated maximum $\delta^{66}\text{Zn}$ values of melts produced by such a carbonate-free
refertilized peridotite is 0.28‰ only (Fig.2b, Supplementary Fig. 3 and Supplementary information,
Note 4). Thus, this "phantom" model, involving pure silicate liquids, will not generate the elevated
$\delta^{66}\text{Zn}$ signatures (0.38±0.03‰) observed in HIMU basalts (see Fig.2b and Supplementary
information, Note 4)."

Lines 195-198: I disagree with the direct use of olivine Ca and Ca/Al ratio, which are both sensitive
to CO₂ content and lithology in the mantle. They should either delete these statement or explore
more on how CO₂ influences olivine Ca and Ca/Al.

**Response:** Revised. Thank you! Please see lines **212-213**.

"Such metasomatized peridotite is enriched by carbonatitic fluids with low SiO₂, Al₂O₃ and
high CaO contents^{16,41}."

Line 198-201: I don't find this sentence informative. This sentence should be modified to show what
specific characteristics point to carbonated melts.

**Response:** We agree. Revised. Thank you! Please see lines **213-216**:

"In addition, the similarity in trace-element patterns, e.g., depletions in K, Rb and Pb relative
to other incompatible elements, between HIMU lavas and high-Mg carbonatitic liquids encapsulated
in diamonds also suggests the involvement of carbonatite metasomatism in the HIMU source¹⁶."

Lines 279-285: The authors should explain here how such heavy Zn can survive the subduction
processes, since CaCO₃ would not be stable in the subduction zone, and it would either melt or be
transformed to other carbonate phases.

**Response:** Done. This is the same question of the first new question of your reviews. Thank you!

See Lines **253-257** in the revised manuscript.

Reviewer #3 (Remarks to the Author):

I have read the authors' responses to those comments carefully, overall I think the problem raised are well addressed. I don't have more questions. My recommendation certainly is publication.

**Response to reviewers' comments in 3rd round:**

A complete list of all the concerns and comments of the reviewers, and the
corresponding response (in blue text), are given below.

Reviewer #3 (Remarks to the Author):

I have read the authors' responses to those comments carefully, overall I think the
problem raised are well addressed. I don't have more questions. My recommendation
certainly is publication.

**Response:** Thank you for your review!
